# Enhanced CO evolution for photocatalytic conversion of $CO_2$ by $H_2O$ over Ca modified $Ga_2O_3$

Rui Pang[1], Kentaro Teramura [1,2 ✉], Masashige Morishita[1], Hiroyuki Asakura [1,2], Saburo Hosokawa[1,2] & Tsunehiro Tanaka[1,2 ✉]

Artificial photosynthesis is a desirable critical technology for the conversion of $CO_2$ and $H_2O$, which are abundant raw materials, into fuels and chemical feedstocks. Similar to plant photosynthesis, artificial photosynthesis can produce CO, $CH_3OH$, $CH_4$, and preferably higher hydrocarbons from $CO_2$ using $H_2O$ as an electron donor and solar light. At present, only insufficient amounts of $CO_2$-reduction products such as CO, $CH_3OH$, and $CH_4$ have been obtained using such a photocatalytic and photoelectrochemical conversion process. Here, we demonstrate that photocatalytic $CO_2$ conversion with a Ag@Cr-decorated mixture of $CaGa_4O_7$-loaded $Ga_2O_3$ and the CaO photocatalyst leads to a satisfactory CO formation rate ($>835$ μmol $h^{-1}$) and excellent selectivity toward CO evolution (95%), with $O_2$ as the stoichiometric oxidation product of $H_2O$. Our photocatalytic system can convert $CO_2$ gas into CO at $>1\%$ $CO_2$ conversion ($>11531$ ppm CO) at ambient temperatures and pressures.

[1] Department of Molecular Engineering, Graduate School of Engineering, Kyoto University, Kyotodaigaku Katsura, Nishikyo-ku, Kyoto 615-8510, Japan. [2] Element Strategy Initiative for Catalysts & Batteries (ESICB), Kyoto University, 1-30 Goryo-Ohara, Nishikyo-ku, Kyoto 615-8245, Japan. ✉email: teramura@moleng.kyoto-u.ac.jp; tanakat@moleng.kyoto-u.ac.jp

Carbon dioxide ($CO_2$) concentrations in the atmosphere have increased drastically over the past few centuries owing to the combustion of carbon-rich fossil fuels such as coal, oil, and natural gas. As a major anthropogenic greenhouse gas, these ever-increasing $CO_2$ emissions are detrimental to the environment and will affect both ecosystems and the global climate[1]. Therefore, there is a critical requirement of mitigating $CO_2$ emissions to achieve sustainable development. Since the pioneering work on the photocatalytic conversion of $CO_2$ into formic acid (HCOOH) and methyl alcohol ($CH_3OH$) over semiconductors reported by Halmann and Inoue et al.[2,3], the photocatalytic conversion of $CO_2$ into other valuable feedstocks at ambient temperatures and pressures has attracted considerable attention from the scientific community as a feasible strategy for $CO_2$ storage and conversion[4–8].

In general, the photocatalytic conversion of $CO_2$ over an excited semiconductor-based catalyst involves three main steps. First, $CO_2$ molecules are adsorbed on the photocatalyst surface[9–11]. Second, the photogenerated electrons react with the adsorbed $CO_2$ species and protons ($H^+$) to yield products such as carbon monoxide (CO), methane ($CH_4$), $CH_3OH$, and HCOOH. Among these possible reduction products, CO is one of the most useful because it is widely combined with $H_2$ to provide synthetic gas for use in many chemical processes, such as methanol synthesis[12,13] and the industrial Fischer–Tropsch process that produce various chemicals and synthetic fuels[14,15]. Third, the products are desorbed from the photocatalyst surface. However, as the $H/H_2$ redox potential ($-0.41$ V vs. NHE at pH 7) is more positive than that for $CO_2/CO$ ($-0.52$ V vs. NHE at pH 7), the generation of $H_2$ from $H^+$ is preferable for the photocatalytic conversion of $CO_2$ into CO, where $H_2O$ acts as the electron donor[16–18]. Moreover, because of the high thermodynamic stability of the linear $CO_2$ molecule, the fixation and activation of $CO_2$ are also immense challenges in the photocatalytic conversion of $CO_2$ by $H_2O$[4,19]. Thus, although various heterogeneous photocatalysts have been reported for the photocatalytic conversion of $CO_2$ into CO with $H_2O$ as the electron donor[20–24], the photocatalytic activity for CO evolution remains limited to a few micromoles, while the photocatalytic conversion rate of $CO_2$ into CO is <0.15%.

Based on the processes involved in the photocatalytic conversion of $CO_2$ described previously, we deduce that the photocatalytic activity of the photocatalyst for $CO_2$ conversion can be improved by increasing $CO_2$ adsorption, charge separation, and product desorption. Due to the fact that $CO_2$ acts as a Lewis acid that bonds easily with Lewis bases[25], many studies have focused on improving $CO_2$ adsorption by modifying the photocatalyst surface with a $CO_2$ adsorbent, such as NaOH[26], amino groups[27], and rare earth species[28], to increase the photocatalytic activity and selectivity for $CO_2$ conversion by $H_2O$. Our group reported that modifying the photocatalyst surface with alkaline earth metals (e.g., Ca, Sr, and Ba) enhanced the conversion of $CO_2$ and the selectivity toward CO evolution[29]. Moreover, we found that a

Ag@Cr core/shell cocatalyst suppresses the backward reaction from CO and $O_2$ to $CO_2$, and enhances the adsorption of $CO_2$, resulting in a highly selective photocatalytic $CO_2$ conversion[30,31].

In this study, we exploited the above techniques and successfully fabricated a Ag@Cr-decorated mixture of $CaGa_4O_7$-loaded $Ga_2O_3$ and CaO photocatalyst, which exhibits a high CO formation rate (>835 μmol $h^{-1}$) per 0.5 g of catalyst, in addition to high selectivity toward CO evolution (>95%) with the stoichiometric production of $O_2$ as the oxidation product of $H_2O$ during the photocatalytic conversion of $CO_2$ by $H_2O$. Approximately 1.2% of the $CO_2$ in the gas phase was transformed into CO (11531 ppm) as a product. The results reported in this study represent almost an order of magnitude higher than most previously published results, as summarized in Supplementary Table 1.

## Results and discussion

**Photocatalytic reduction of $CO_2$ by $H_2O$.** Table 1 shows the formation rates of CO, $H_2$, and $O_2$, selectivity toward CO evolution, and the balance between consumed electrons and holes over the bare $Ga_2O_3$, Ag-modified $Ga_2O_3$ (Ag/$Ga_2O_3$), Ag@Cr-modified $Ga_2O_3$ (Ag@Cr/$Ga_2O_3$), and Ag@Cr-modified Ca-loaded $Ga_2O_3$ (Ag@Cr/$Ga_2O_3$_Ca) photocatalysts during the photocatalytic conversion of $CO_2$ by $H_2O$. No liquid products were detected in the reaction solutions in these photocatalytic systems, and $H_2$, $O_2$, and CO were detected as gaseous products. As no reduction products other than $H_2$ and CO were generated, the selectivity toward CO evolution and the balance between the consumed electrons and holes were calculated as follows:

$$\text{Selectivity toward CO evolution } (\%) = 2R_{CO}/(2R_{CO} + 2R_{H2}) \times 100, \quad (1)$$

$$\text{Consumed } e^-/h^+ = (2R_{CO} + 2R_{H2})/4R_{O2}, \quad (2)$$

where $R_{CO}$ and $R_{H2}$ represent the formation rates of CO and $H_2$, respectively. If $H_2O$ acts as an electron donor, the value of $e^-/h^+$ should be equal to 1.

We obtained stoichiometric amounts of $H_2$ and CO as reduction products in addition to $O_2$ as the oxidation product, indicating that $H_2O$ serves as the electron donor. Bare $Ga_2O_3$ exhibited a particularly low selectivity toward CO evolution (4%) as the electrons generated by charge transfer were not consumed in the reduction of $CO_2$, but rather in the production of $H_2$ from $H^+$. Modifying $Ga_2O_3$ with a Ag cocatalyst enhanced the selectivity toward CO evolution (29%); however, this was not sufficient to obtain a selectivity >50%. In contrast, we succeeded in the selective photocatalytic conversion of $CO_2$ by $H_2O$ over Ag@Cr/$Ga_2O_3$. A relatively high CO formation rate (499.6 μmol $h^{-1}$) was achieved with 77% selectivity toward CO evolution. The photocatalytic reaction for the conversion of $CO_2$ by $H_2O$ over Ag@Cr/$Ga_2O_3$ and Ag@Cr/$Ga_2O_3$_Ca was

**Table 1 Photocatalytic conversions of $CO_2$ by $H_2O$ using various photocatalysts.**

| Catalyst | Formation rates of products (μmol $h^{-1}$) | | | Selec. toward CO (%) | Consumed $e^-/h^+$ |
|---|---|---|---|---|---|
| | $H_2$ | $O_2$ | CO | | |
| Bare $Ga_2O_3$ | 240.9 | 122.8 | 9.8 | 4 | 1.02 |
| Ag/$Ga_2O_3$ | 248.3 | 172.7 | 102.1 | 29 | 1.01 |
| Ag@Cr/$Ga_2O_3$ | 148.5 | 316.4 | 499.6 | 77 | 1.02 |
| Ag@Cr/$Ga_2O_3$_Ca | 176.5 | 448.2 | 794.2 | 82 | 1.08 |

Photocatalyst powder: 0.5 g, reaction solution volume: 1.0 L, additive: 0.1 M $NaHCO_3$, $CO_2$ flow rate: 30 mL $min^{-1}$, light source: 400-W high-pressure Hg lamp.

carried out for at least four times, and errors in the product formation rates ($H_2$, $O_2$, and CO) were smaller than 5%. Controlling both, the bulk and surface of the photocatalyst, is highly important for achieving a considerably high CO formation rate and selectivity toward CO evolution. We found that the amount of Ca species significantly affected the $H_2$ and CO formation rates (for the product formation rates and selectivity over various Ag@Cr/$Ga_2O_3$_Ca photocatalysts see Supplementary Fig. 1). The formation rate of CO increased first and then decreased as the Ca content increased (Supplementary Fig. 1a–g). In contrast, the formation rate of $H_2$ over the Ag–Cr/$Ga_2O_3$_Ca_$x$ samples increased monotonically with increasing amount of Ca species. The Ag–Cr/$CaGa_4O_7$ photocatalyst was only active for $H_2$ evolution derived from water splitting (Supplementary Fig. 1h). The Ag@Cr/$Ga_2O_3$_Ca photocatalyst exhibited the highest CO formation rate (794.2 µmol h$^{-1}$), and the selectivity toward CO evolution was approximately 82%. Additionally, CO production from the photocatalytic conversion of $CO_2$ after photoirradiation for 15 h over Ag@Cr/$Ga_2O_3$_Ca was more stable than that over Ag@Cr/$Ga_2O_3$ (for the product formation rates for 15 h see Supplementary Fig. 2), which indicates that the presence of Ca species is not only beneficial for improving the photocatalytic activity and selectivity, but also for improving stability during the photocatalytic conversion of $CO_2$ to CO.

Various control experiments were carried out to confirm the source of CO during the photocatalytic conversion of $CO_2$ by $H_2O$, the results of which are shown in Supplementary Fig. 3. We did not detect any appreciable amounts of products under dark conditions or in the absence of a photocatalyst. In addition, $H_2$ was the main product formed when Ar gas was used instead of $CO_2$ or in the absence of $NaHCO_3$. The control experiments confirmed that the evolved CO originated from the $CO_2$ gas introduced into the samples and not from carbon contaminants.

**Photocatalyst characterization**. The actual amounts of the Ca species loaded into $Ga_2O_3$ at different $CaCl_2$ concentrations were measured using inductively coupled plasma optical emission spectrometry (ICP-OES) (Supplementary Table 2). We found that almost all the Ca species were loaded into the $Ga_2O_3$ photocatalyst when the $CaCl_2$ concentration was <0.001 mol L$^{-1}$. However, not all the Ca species could be loaded into $Ga_2O_3$ at higher $CaCl_2$ concentrations. Note that even when no $CaCl_2$ was added during the preparation of $Ga_2O_3$, trace amounts of Ca were detected in $Ga_2O_3$, which is likely due to Ca impurities present in the experimental vessels or precursor reagents. Hereinafter, we refer to the Ca-loaded $Ga_2O_3$ photocatalysts as $Ga_2O_3$_Ca_$x$ ($x$ = 0.32, 0.62, 1.1, 1.6, 2.1, 3.3 mol%) based on the Ca/Ga molar ratio determined by ICP-OES. Figure 1a shows the X-ray diffraction (XRD) patterns of the bare $Ga_2O_3$, $Ga_2O_3$_Ca_$x$, and $CaGa_4O_7$ photocatalysts. As indicated, gradual changes in the diffraction peaks assigned to the (020), (311), (400), (002), and (330) facets of $CaGa_4O_7$ (JSPDS 01-071-1613) were observed as the amount of Ca species was increased. In general, a high Ca loading is favorable for the formation of $CaGa_4O_7$. We observed no distinct shifts in the diffraction peaks for the $Ga_2O_3$_Ca_$x$ samples compared with those of bare $Ga_2O_3$. As the ionic radius of $Ca^{2+}$ (0.099 nm)[32] is larger than that of $Ga^{3+}$ (0.062 nm)[33], the unshifted XRD peaks imply that $Ca^{2+}$ does not act as a dopant in the bulk $Ga_2O_3$ lattice. However, there was a clear increase in the peak intensity at $2\theta$ = 30.1° and an apparent decrease in that at $2\theta$ = 30.5° with increasing amount of Ca species (Fig. 1b), which are possibly ascribed to the formation of $CaGa_4O_7$ species on $Ga_2O_3$. The increased intensity of the Ca 2p X-ray photoelectron spectroscopy (XPS) peak (Fig. 1c) also indicates that the amount

of Ca species on the $Ga_2O_3$ surface increased with increasing Ca levels. In addition, the XPS peak locations in the Ca 2p spectra of the $Ga_2O_3$_Ca_$x$ photocatalysts are similar to those of $CaGa_4O_7$, but different from those of CaO. The Ca 2p XPS profiles suggest that a thin $CaGa_4O_7$ layer forms on the $Ga_2O_3$ surface and that the amount of $CaGa_4O_7$ increases as the amount of Ca is increased. We further confirmed the morphological changes in the $Ga_2O_3$_Ca sample by field-emission scanning electron microscopy (SEM), as shown in Fig. 1d. Both ends of the $Ga_2O_3$ nanoparticles gradually sharpened and their surfaces became smoother as the amount of Ca species increased, especially when the Ca amount was higher than 1.1 mol%. This smoothing of the $Ga_2O_3$ surfaces with increasing Ca/Ga molar ratio resulted in a decrease in the Brunauer–Emmett–Teller (BET) specific surface area of $Ga_2O_3$_Ca_$x$ (Supplementary Fig. 4), which is attributable to the modification of $CaGa_4O_7$, as we confirmed from the XRD patterns and the XPS results that a $CaGa_4O_7$ layer was formed on the $Ga_2O_3$ surface.

The close linkage between $CaGa_4O_7$ and $Ga_2O_3$ on the $Ga_2O_3$ surface was confirmed by field-emission transmission electron microscopy (TEM) and high-resolution TEM (HRTEM) (Fig. 2). The marked lattice spacings (0.296 and 0.255 nm) in Fig. 2b correspond to the (130) and (111) planes of $CaGa_4O_7$ and $Ga_2O_3$, respectively. The core–shell-structured Ag@Cr cocatalyst was successfully loaded onto the $Ga_2O_3$_Ca surface using the photodeposition method (Fig. 2c, d), as reported previously by us[31].

**Role of the Ca species**. Figure 3 shows the Fourier transform infrared (FTIR) spectra of the $CO_2$-adsorbed samples after introducing $CO_2$ at ~0.2 Torr. When $CO_2$ was introduced into the $Ga_2O_3$ sample, three absorbance peaks were observed at 1634, 1432, and 1225 cm$^{-1}$, which can be ascribed to asymmetric $CO_3$ stretching vibrations [$\nu_{as}(CO_3)$], symmetric $CO_3$ stretching vibrations [$\nu_s(CO_3)$] of monodentate bicarbonate species (m-$HCO_3$-Ga), and OH deformation vibrations [$\delta(OH)$], respectively[34–36]. The absorbance peaks at 1699 and 1636 cm$^{-1}$ for the $CO_2$-adsorbed CaO sample can be attributed to bridging carbonate stretching and asymmetric $CO_3$ stretching vibrations [$\nu_{as}(CO_3)$] of the bicarbonate species, respectively. The broad structureless absorbance peaks between 1480 and 1318 cm$^{-1}$ can be attributed to the symmetric and asymmetric $CO_3$ stretching of unidentate carbonate, as well as the symmetric $CO_3$ stretching [$\nu_s(CO_3)$] of bicarbonate[37–41]. When the $Ga_2O_3$ surface was modified with a small amount of Ca species, absorbance peaks attributable to $CO_2$ adsorption by both $Ga_2O_3$ and CaO were observed after $CO_2$ was introduced into the $Ga_2O_3$_Ca_1.1 sample. However, when the $Ga_2O_3$ surface was modified with large amounts of Ca species, the absorbance peaks attributed to $CO_2$ adsorption on $Ga_2O_3$ had low intensity and mainly corresponded to the broad peaks derived from the adsorption of $CO_2$ on $CaGa_4O_7$. Supplementary Fig. 5 shows the FTIR spectra of $CO_2$-adsorbed $Ga_2O_3$, $Ga_2O_3$_Ca_1.1, $Ga_2O_3$_Ca_3.3, and $CaGa_4O_7$ samples after introducing the same amount of $CO_2$ at various pressures in the 0.1–40.0 Torr range. $CO_2$ was adsorbed significantly more on the $Ga_2O_3$_Ca_1.1 surface than on the $Ga_2O_3$ surface due to its adsorption at both Ga and Ca sites. However, the $CaGa_4O_7$ surface was not conducive to $CO_2$ adsorption; therefore, $CO_2$ adsorbed less onto the $Ga_2O_3$_Ca_3.3 surface than the $Ga_2O_3$_Ca_1.1 surface.

Figure 4 shows the FTIR spectra of the adsorbed $CO_2$ species on $Ga_2O_3$_Ca_1.1 after different durations of photoirradiation. As the photoirradiation time increased from 0 to 106 h, the bands at 1225 [$\delta(OH)$-Ga] and 1408 cm$^{-1}$ [$\nu_s(CO_3)$-Ca] decreased and

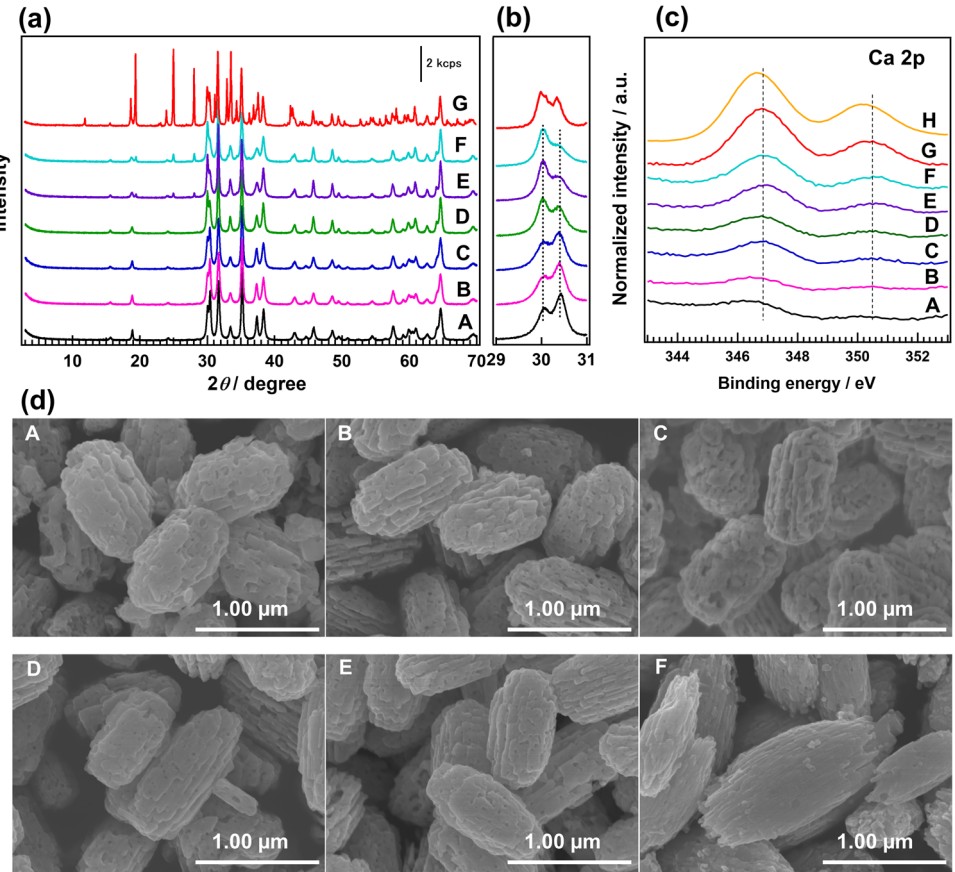

**Fig. 1 Photocatalyst characterization. a** X-ray diffractograms; **b** enlarged X-ray diffractograms at $2\theta = 29–31°$; **c** Ca 2p X-ray photoelectron spectroscopy profiles; and **d** field-emission scanning electron microscopy images of **A** bare $Ga_2O_3$; $Ga_2O_3\_Ca\_x$ with a Ca/Ga molar ratio $x$ of **B** 0.32 mol%, **C** 0.62 mol%, **D** 1.1 mol%, **E** 2.1 mol%, and **F** 3.3 mol%; **G** $CaGa_4O_7$ (in **a**, **b**), and **H** CaO (in **c**).

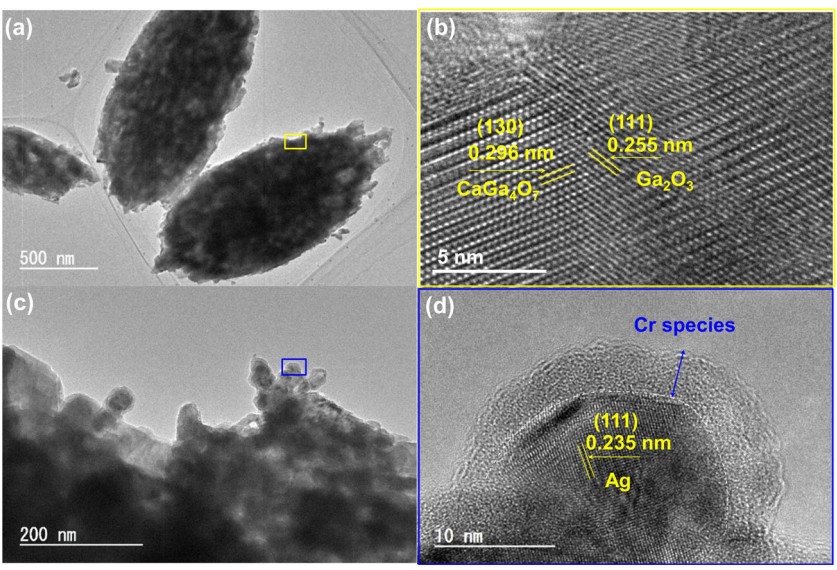

**Fig. 2 Transmission electron microscopy (TEM) images of the photocatalysts.** TEM images of **a** $Ga_2O_3\_Ca$ and **c** $Ag@Cr/Ga_2O_3\_Ca$. High-resolution TEM images of **b** $Ga_2O_3\_Ca$ and **d** $Ag@Cr/Ga_2O_3\_Ca$. Note that **b**, **d** are enlarged TEM images of the marked areas in **a**, **c** indicated by yellow and blue boxes, respectively.

vanished after 104 h. At the same time, new bands gradually appeared at 1581, 1388, and 1353 cm$^{-1}$ (asymmetric $CO_2$ stretching [$\nu_{as}(CO_2)$], CH deformation [$\delta(CH)$], and symmetric $CO_2$ stretching [$\nu_s(CO_2)$] assigned to formate species (HCOO–Ga/Ca), respectively)[34–36]. As the photoirradiation

continued, the formate species were consumed and gaseous CO (fundamental vibration band at 2143 cm$^{-1}$)[42] was formed simultaneously. This result indicates that the bicarbonate species is the intermediate during the photocatalytic conversion of $CO_2$, and the formates transform into CO with photoirradiation, which

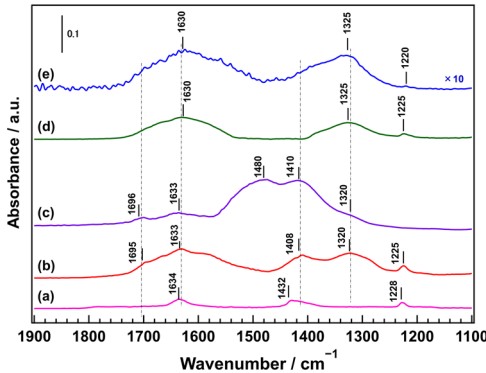

**Fig. 3 Fourier transform infrared spectra of $CO_2$ adsorption.** $CO_2$ adsorbed on: **a** $Ga_2O_3$, **b** $Ga_2O_3\_Ca\_1.1$, **c** CaO, **d** $Ga_2O_3\_Ca\_3.3$, and **e** $CaGa_4O_7$ after introducing ~0.2 Torr of $CO_2$.

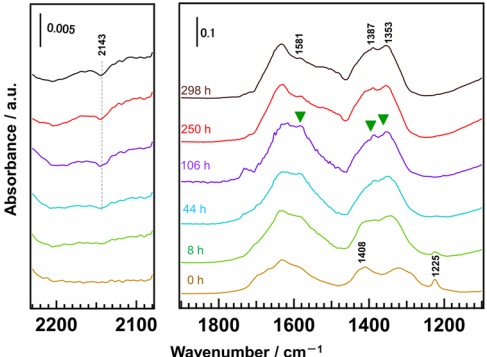

**Fig. 4 FTIR spectra of $CO_2$ adsorption under photoirradiation.** Difference FTIR spectra of the adsorbed $CO_2$ species on $Ga_2O_3\_Ca\_1.1$ under photoirradiation for different hours. ~2.0 Torr of $CO_2$ was introduced into the instrument.

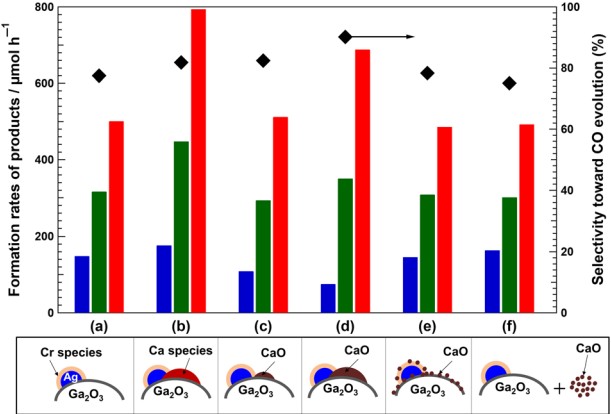

**Fig. 5 Product formation rates and selectivity.** Rates of formation of $H_2$ (blue bars), $O_2$ (green bars), and CO (red bars), as well as selectivity toward CO evolution (black diamonds) for various photocatalysts: **a** $Ag@Cr/Ga_2O_3$; **b** $Ag@Cr/Ga_2O_3\_Ca\_1.1$; **c** $Ag@Cr/(1.1\ mol\%CaO/Ga_2O_3)$, namely $Ga_2O_3$ physically mixed with 1.1 mol% of CaO by grinding before loading the Ag@Cr cocatalyst; **d** $Ag@Cr/(30\ mol\%CaO/Ga_2O_3)$, which is similar to **c** except for using 30 mol% of CaO; **e** $Ag@Cr/Ga_2O_3 + 30\ mol\%$ CaO, namely $Ag@Cr/Ga_2O_3$ physically mixed with 30 mol% of CaO by grinding; and **f** $Ag@Cr/Ga_2O_3$ and 30 mol% of CaO without mixing before adding into the reaction solution. Schematic structures of the photocatalysts are shown at bottom. Photocatalyst powder: 0.5 g, reaction solution volume: 1.0 L, additive: 0.1 M $NaHCO_3$, $CO_2$ flow rate: 30 mL min⁻¹, light source: 400 W high-pressure Hg lamp.

is consistent with our previous results[43,44]. It is worth mentioning that in addition to the presence of intermediate species on the $Ga_2O_3$ surface ($[\delta(OH)–Ga]$), the modification by Ca species further increased the amount of intermediate on $Ga_2O_3\_Ca\_1.1$. As the photocatalytic conversion of $H^+$ into $H_2$ and the conversion of $CO_2$ into CO are two competing processes in an aqueous solution, the high adsorption of $CO_2$ at the base site leads to high photocatalytic activity and selectivity toward CO evolution during the photocatalytic conversion of $CO_2$ by $H_2O$.

In order to demonstrate that the presence of CaO on the $Ga_2O_3$ surface enhances the photocatalytic activity and selectivity during the photocatalytic conversion of $CO_2$ into CO, we investigated the photocatalytic performance during the conversion of $CO_2$ by $H_2O$ over various $Ag@Cr/CaO/Ga_2O_3$ photocatalysts, the results of which are shown in Fig. 5. We found that the Ag@Cr/ $Ga_2O_3\_Ca\_1.1$ photocatalyst (with a low amount of CaO generated on the $Ga_2O_3$ surface) significantly enhanced the rate of CO formation during the photocatalytic conversion of $CO_2$ by $H_2O$ compared with bare $Ag@Cr/Ga_2O_3$ (Fig. 5a, b). However, no significant change in the rate of CO formation and selectivity toward CO evolution was observed for the sample labeled "$Ag@Cr/(1.1\ mol\%CaO/Ga_2O_3)$" (in which 1.1 mol% CaO was physically loaded onto $Ga_2O_3$ by grinding before loading Ag@Cr cocatalyst onto the $CaO/Ga_2O_3$ surface) as compared to bare $Ga_2O_3$ (Fig. 5c). Because uncalcined CaO-loaded $Ga_2O_3$ easily dissolves in $H_2O$, we increased the CaO loading on the $Ga_2O_3$ surface to 30 mol% using the same grinding method (labeled "$Ag@Cr/(30\ mol\%CaO/Ga_2O_3)$"), which resulted in an increased rate of CO formation and a decrease in $H_2$ formation (Fig. 5d).

However, no improvement in photocatalytic activity and selectivity was observed when 30 mol% CaO was mixed with the prepared $Ag@Cr/Ga_2O_3$ and ground together (Fig. 5e) or when they were directly mixed in the reaction solution (Fig. 5f). These results clearly reveal that the addition of CaO on the $Ga_2O_3$ surface enhances the rate of CO formation and suppresses that of $H_2$ during the photocatalytic conversion of $CO_2$ by $H_2O$. In addition, the tight junction between $Ga_2O_3$, CaO, and the Ag@Cr cocatalyst is crucial for the superior photocatalytic activity and selectivity of the photocatalyst for the conversion of $CO_2$ into CO. In our previous work, we confirmed that Ag acts as an active site while the $Cr(OH)_3 \cdot H_2O$ layer exterior to the Ag core increases $CO_2$ adsorption[30,31]. Hence, the Ag@Cr cocatalyst should be loaded at the $CaO/Ga_2O_3$ interface in order to facilitate contact between the CaO-adsorbed $CO_2$ species and the Ag active sites.

Notably, although $CaGa_4O_7$ exhibited high selectivity toward $H_2$ evolution, the $H_2$ formation rate for $CaGa_4O_7$ was significantly lower than that for $Ga_2O_3\_Ca\_3.3$ (for the product formation rates over $Ag$-$Cr/Ga_2O_3\_Ca\_3.3$ see Supplementary Fig. 6). This indicates that the presence of $CaGa_4O_7$ on the $Ga_2O_3$ surface enhances the overall photocatalytic efficiency during the photocatalytic reaction, including $CO_2$ conversion and water splitting. The Mott–Schottky plot (Supplementary Fig. 7) and the absorption spectra converted from the diffuse reflectance spectra using the Kubelka–Munk equation (Supplementary Fig. 8) enabled us to estimate the conduction band (CB) and valence band (VB) positions of $Ga_2O_3$, $Ga_2O_3\_Ca\_0.62$, and $CaGa_2O_7$, as shown in Supplementary Fig. 9. Since the CB and VB of $Ga_2O_3$ are both more positive than those of $CaGa_4O_7$, a heterojunction system between $Ga_2O_3$ and $CaGa_4O_7$ can be formed that greatly improves the spatial separation efficiency of the photogenerated carriers[45]. Therefore, $CaGa_4O_7/Ga_2O_3$ exhibited a much higher photocatalytic efficiency than bare $Ga_2O_3$ and $CaGa_4O_7$.

We expect that by exploiting the high $CO_2$ adsorption of CaO and the high photocatalytic efficiency of $CaGa_4O_7/Ga_2O_3$, we can

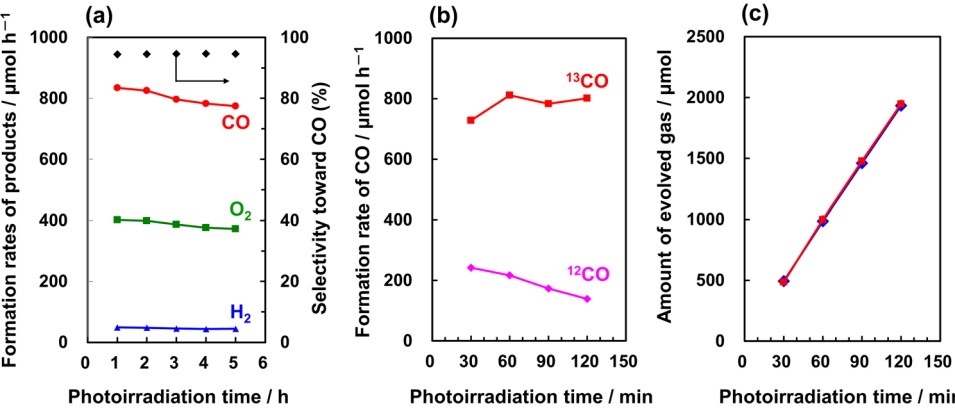

**Fig. 6 CO, O$_2$, and H$_2$ formation data. a** Formation rates of H$_2$ (blue triangles), O$_2$ (green squares), and CO (red circles), and selectivity toward CO evolution (black diamonds) for the photocatalytic conversion of CO$_2$ by H$_2$O; **b** $^{12}$CO and $^{13}$CO detected by MS ($m/z = 28$ and 29) from the photocatalytic conversion of $^{13}$CO$_2$ by H$_2$O; **c** CO time-course as determined by MS (red squares) and GC (blue diamonds) for the photocatalytic conversion of $^{13}$CO$_2$ by H$_2$O.

further improve the photocatalytic activity and selectivity of the photocatalyst to maximize the conversion of CO$_2$ into CO by H$_2$O. Figure 6a shows the formation rates of H$_2$, O$_2$, and CO during the photocatalytic conversion of CO$_2$ by H$_2$O for the Ga$_2$O$_3$\_Ca\_3.3 photocatalyst physically mixed with 30 mol% of CaO and Ag@Cr as the cocatalyst. As indicated, a high formation rate of CO (>835 μmol h$^{-1}$) was achieved, in addition to an excellent selectivity toward CO evolution (>95%), with a stoichiometric amount of evolved O$_2$. Both $^{12}$CO and $^{13}$CO were detected using quadrupole mass spectrometry (MS), and the peaks at $m/z = 28$ and $m/z = 29$ were located at the same positions as those detected by gas chromatography (GC) during the photocatalytic conversion of $^{13}$CO$_2$ (for the isotopic lead experiments see Supplementary Fig. 10). Indeed, our results indicate that the detected $^{12}$CO was produced from the reduction of $^{12}$CO$_2$ derived from the NaHCO$_3$ additive in the solution[43]. As shown in Fig. 6b, with the consumption of $^{12}$CO$_2$ derived from NaHCO$_3$, the amount of generated $^{12}$CO gradually decreased, while the $^{13}$CO content increased under continuous bubbling of $^{13}$CO$_2$. The total amounts of $^{13}$CO and $^{12}$CO detected by MS were consistent with the amount of CO detected by GC (Fig. 6c), which indicates that the CO was generated as the reduction product of either CO$_2$ introduced in the gas phase or from NaHCO$_3$, rather than from any organic contaminants on the photocatalyst surface. The converted concentration of CO based on the CO formation rate was found to be 11,531 ppm, indicating that ~1.2% of CO$_2$ in the gas phase was transformed into CO (see Supplementary Information for the calculation details. The actual amounts of CO detected are shown in Supplementary Movie 1).

In our previous work, we had found that basic oxides and hydroxides such as Cr(OH)$_3$[31], SrO[44], and rare earth (RE) hydrates and oxides[28] function as good CO$_2$ storage materials by generating the corresponding (hydroxy)carbonate compounds (e.g., Cr(OH)$_x$(CO$_3$)$_y$ and RE$_2$(OH)$_{2(3−x)}$(CO$_3$)$_x$), and they improve the photocatalytic activity and selectivity toward CO evolution. Now, we propose a possible mechanism for the photocatalytic conversion of CO$_2$ by H$_2$O over Ag@Cr/CaO/CaGa$_2$O$_7$/Ga$_2$O$_3$, as shown in Fig. 7. During the photocatalytic conversion of CO$_2$ in an aqueous solution of NaHCO$_3$, the Cr(OH)$_3$·H$_2$O and CaO species that are in close contact with Ag particles easily form (hydroxy)carbonate species (named M(OH)$_x$(CO$_3$)$_y$, M=Cr or Ca)[31], which greatly increase the concentration of CO$_2$-related species around the Ag active sites, thereby improving selectivity for the photocatalytic conversion of CO$_2$ into CO instead of water splitting. On the other hand, the Ga$_2$O$_3$/CaGa$_4$O$_7$ heterojunction improves the efficiency for

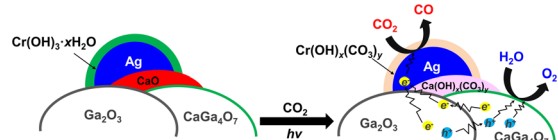

**Fig. 7 A plausible reaction mechanism.** Schematic illustration of the mechanism for the photocatalytic conversion of CO$_2$ into CO over Ag@Cr/CaO/CaGa$_4$O$_7$/Ga$_2$O$_3$.

spatial separation of the photogenerated carriers, which also increases the photocatalytic activity for the conversion of CO$_2$ into CO. Moreover, while the Cr(OH)$_3$·xH$_2$O shell outside the Ag particle can be oxidized to Cr$^{6+}$ and dissolve into the solution during the photocatalytic conversion of CO$_2$[46], the presence of CaO around the Ag active site compensates for the reduced activity from the dissolution of Cr species. As a result, Ag@Cr/Ga$_2$O$_3$\_Ca is photocatalytically much more stable than Ag@Cr/Ga$_2$O$_3$.

Herein, we reported the photocatalytic conversion of CO$_2$ using a Ag@Cr/CaO/CaGa$_4$O$_7$/Ga$_2$O$_3$ photocatalyst, in which a satisfactory CO formation rate (>835 μmol h$^{-1}$) and an excellent selectivity toward CO evolution (95%) were achieved with the stoichiometric production of O$_2$ as the oxidation product of H$_2$O. Through the use of various characterization techniques, we found that the CaO and CaGa$_4$O$_7$ formed on the Ga$_2$O$_3$ surface improved the adsorption of CO$_2$ at basic sites in addition to enhancing the total photocatalytic efficiency. In addition, the physical mixing of CaGa$_4$O$_7$/Ga$_2$O$_3$ with CaO was a particularly simple and convenient technique for exploiting the high CO$_2$ adsorption ability of CaO and the high photocatalytic efficiency of CaGa$_4$O$_7$/Ga$_2$O$_3$. These results are of particular interest, considering that previously, only insufficient amounts of CO$_2$ reduction products were produced during artificial photosynthesis.

## Methods

Ca-modified Ga$_2$O$_3$ (Ga$_2$O$_3$\_Ca) was prepared using the ammonia precipitation method reported by Sakata et al.[47]. In this method, Ga(NO$_3$)$_3$·$n$H$_2$O (12.6 g) was dissolved in 200 mL of deionized water or CaCl$_2$ solution in ultrapure water at various concentrations. Hydroxylation was carried out by dripping an ammonium hydroxide solution until the pH level reached 9.1. The obtained hydroxides were centrifuged and dried overnight. The Ga$_2$O$_3$\_Ca sample was obtained by calcining the precursor at 1273 K for 10 h. Ag@Cr/Ga$_2$O$_3$\_Ca was synthesized using the photodeposition method reported in our previous work[30]. In this method, the as-prepared Ga$_2$O$_3$\_Ca powder (1.0 g) was dispersed in ultrapure water (1.0 L) containing the necessary amounts of silver nitrate (AgNO$_3$) and chromium (III) nitrate

(Cr(NO$_3$)$_3$). The suspension was purged with Ar gas and irradiated under a 400 W high-pressure Hg lamp with Ar gas flowing for 1.0 h, followed by filtration and drying at room temperature (~298 K). The Ag/Ga and Cr/Ga molar ratios were both 1.0 mol%.

**Characterization**. The as-prepared Ga$_2$O$_3$_Ca samples were characterized using the following techniques: XRD (Model: Multiflex, Rigaku Corporation, Japan) with Cu Kα radiation (λ = 0.154 nm); XPS (Model: ESCA 3400, Shimadzu Corporation, Japan) with Mg Kα radiation; SEM (Model: SU-8220, Hitachi High-Technologies Corporation, Japan); TEM (Model: JEM-2100F, JEOL Ltd, Japan); and UV–Visible spectroscopy (V-650, JASCO) with an integrated sphere accessory. The BET surface areas of the photocatalyst samples were determined from their N$_2$-adsorption isotherms at 77 K using a volumetric gas-adsorption measuring instrument (Model: BELSORP-miniII, MicrotracBEL Corp. (formerly BEL Japan, Inc.), Japan). Prior to these measurements, each sample was evacuated at 473 K for 1 h using a sample pretreatment system (Model: BELPREP-vacII, MicrotracBEL Corp. (formerly BEL Japan, Inc.), Japan). ICP-OES (Model: iCAP7400, Thermo Fisher Scientific, USA) was used to determine the actual amounts of Ca modified on the Ga$_2$O$_3$ surface. The FTIR spectra of the adsorbed carbon species were recorded using an FTIR spectrometer (Model: FT/IR-4700, JASCO International Co., Ltd., Japan) equipped with a mercury–cadmium–tellurium (MCT) detector and cooled with liquid N$_2$ in the transmission mode at 303 K. Each sample (~30 mg) was pressed into a wafer (diameter: 10 mm) and introduced into the instrument in a cylindrical glass cell with calcium fluoride (CaF$_2$) windows. The wafer was evacuated at 673 K for 30 min before being examined, followed by treatment with O$_2$ at ~40 Torr for 30 min, after which the wafer was evacuated for 30 min and cooled to 303 K. The data for each FTIR spectrum were obtained from 128 scans with a resolution of 4 cm$^{-1}$. The energy gap of the band structure and flat band potential of the Ga$_2$O$_3$_Ca samples were determined using the Davis–Mott and Mott–Schottky equations, respectively; the experimental details are provided in the Supplementary Information.

**Photocatalytic reaction**. The photocatalytic reduction of CO$_2$ was carried out using a flow system with an inner irradiation-type reaction vessel. The synthesized photocatalyst (0.5 g) was dispersed in ultrapure water (1.0 L) containing 0.1 M sodium bicarbonate (NaHCO$_3$). The CO$_2$ was bubbled into the solution at a flow rate of 30 mL min$^{-1}$. The suspension was illuminated using a 400 W high-pressure Hg lamp with a quartz filter, and the assembly was connected to a water-cooling system. The amounts of evolved H$_2$ and O$_2$ were detected using a gas chromatography system fitted with a thermal conductivity detector (TCD-GC, Model: GC-8A, Shimadzu Corporation, Japan) and a 5A molecular sieve (MS 5A) column, and Ar was used as the carrier gas. The amount of evolved CO was analyzed using a gas chromatography system fitted with a flame ionization detector (FID-GC, Model: GC-8A, Shimadzu Corporation, Japan), a methanizer, and a ShinCarbon ST column, and N$_2$ was used as the carrier gas. High-performance liquid chromatography (Model: LC-4000, JASCO, USA) was used to detect the presence of liquid products.

In the isotope experiment, $^{12}$CO$_2$ was replaced by $^{13}$CO$_2$. The formation rates of H$_2$, O$_2$, $^{12}$CO, and $^{13}$CO under photoirradiation were detected using a quadrupole mass spectrometer (BELMASS, Microtrac BEL) combined with a TCD-GC detector.

## Data availability

The datasets generated during and/or analysed during the current study are available in the [figshare] repository, [https://figshare.com/s/84a5d675a273e507fb55 and/or https://doi.org/10.6084/m9.figshare.12927422].

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

## Acknowledgements
This study was partially supported by a Grant-in-Aid for Scientific Research on Innovative Areas, "All Nippon Artificial Photosynthesis Project for Living Earth" [grant number 2406] of the Ministry of Education, Culture, Sports, Science, and Technology (MEXT) of Japan and the Program for Element Strategy Initiative for Catalysts & Batteries (ESICB) [No. JPMXP0112101003], commissioned by the MEXT of Japan.

## Author contributions
R.P. and K.T. designed the research. R.P. prepared the photocatalyst powder and conducted XRD, BET, XPS, SEM, TEM, ICP-OES, UV–Vis, and FTIR studies, electrochemical measurements, and the photocatalytic $CO_2$-conversion experiments. M.M. carried out the isotopic labeling experiment. R.P., K.T., M.M., H.A., S.H., and T.T. discussed the results. R.P. and K.T. wrote the manuscript with contributions from the other authors.

## Competing interests
The authors declare no competing interests.
