## [Peer Review File · Communications Chemistry]

Reviewers' comments:

Reviewer #1 (Remarks to the Author):

The manuscript by Pang et al. reported that the photocatalytic reduction of CO₂ over an Ag-modified Cr-decorated mixture of CaGa₄O₇/Ga₂O₃ and CaO exhibits an excellent CO formation with good selectivity, and the conversion of CO₂ in water can be up to 1% without any sacrificial agent. The materials are properly characterized, the number and quality of figures and tables is adequate. The paper is an important advance and of good quality, making it suitable for publication in Communications Chemistry. However, I believe there are some points that need to be addressed by the authors before acceptance.

1. For comparison, please summarize some representative results of photocatalytic systems towards CO₂ reduction with H₂O in Table.
2. The Ca species on the Ga₂O₃ surface were in form of CaGa₄O₇ by XRD and XPS, while CaO was found by FTIR when the modification amount of Ca was low. Please provide the Ca 2p XPS of CaO and the FTIR spectra of the CO₂-adsorbed on all Ga₂O₃_Ca_x samples and try to illustrate the existing form of Ca and what roles two Ca species play for the high conversion and selectivity during the photocatalytic CO₂ reduction.
3. The specific compose of CaGa₄O₇/Ga₂O₃ photocatalyst physically mixed with 30 mol% of CaO with Ag-Cr as the cocatalyst should be provided to gain deeper insights into the catalysis. 30 mol% of CaO are the optimum loading or not?
4. Is it possible that H₂ and O₂ are generated from the over-all water splitting?
5. It's known that CaO reacts into Ca(OH)₂ in water, which is responsible for the high CO₂ adsorption. And I find that the formation rate of CO drops gradually in Fig.4(a). It means that the photocatalyst is not stable and/or the consumption of CaO is necessary for the photocatalytic CO₂ reduction. The authors should illustrate this clearly.
6. Energy band structure of CaGa₄O₇/Ga₂O₃ should be provided to clarify their justification for high efficiencies on photocatalytic CO₂ reduction.

Reviewer #2 (Remarks to the Author):

The motivation of the work was not clearly stated. Why did authors select a Ag-modified Cr-decorated mixture of CaGa₄O₇-loaded Ga₂O₃ and CaO photocatalyst? The statement should be clearly described in the paragraphs for introduction. Currently, it is hard to capture what is the motivation of the work, although the addition of Ca or CaO layer is the key to the works.

To make readers understand the principle of CO₂ reduction, it is necessary to suggest the mechanism of charge transfer with schematic illustrations and a band diagram.

The amount of CO₂ adsorbed on each catalyst surface should be measured to confirm the enhanced CO₂ adsorption with the formation of CaO layer. The BET areas measured with N₂ gas decrease with increasing Ca but the CO₂ amount adsorbed should increase, which should be experimentally confirmed.

Reviewer #3 (Remarks to the Author):

The paper from Pang et al describes a material -- Ag@Cr/Ga₂O₃_Ca -- with a remarkable CO₂ photoreduction activity to CO. The production of almost 1 mmol/g.h of CO is impressive, and it is higher than most of the recent papers about CO₂ reduction. The paper has an important feature: this is one of the only papers I've recently read that takes care of the oxidative reaction related to

CO₂ reduction, i.e., O₂ evolution. In that sense, they could explore this data better. Fig 1 and Suppl Fig 1 show the production of CO, H₂, and O₂, comparatively, but it is hard to compare if the stoichiometry is correct or not. For instance, in Fig 1 for Ag@Cr/Ga₂O₃_Ca sample, it seems to me that CO production is around 0.8 mmol/h, H₂ is 0.2 mmol/h, and O₂ is 0.4 mmol/h. Since it is expected that CO₂ → CO produces 1/2 O₂, as well as H₂O → H₂ + 1/2 O₂, I would expect that the reaction would produce at least 0.5 mmol/h. Naturally, this indicates that probably this is a statistical variation question, but since the authors did not provide error bars for their experiments, it can not be confirmed.

A second question about their experiments is the power of the light source. They report using a 4,000 W high-pressure Hg lamp with a Quartz filter (line 268). It is significantly higher than typically reported in similar papers. The spectrum of these lamps includes deep-UV contributions, which can reduce CO₂ with no aid of catalysts. Please give more information about the lamp spectrum. See also JACS dx.doi.org/10.1021/ja304930t, where the authors have discussed this reaction as well as the influence of a basic solid in increasing CO₂ reduction. Therefore, the authors should provide a "blank" experiment, i.e., the measurement of CO evolution in identical conditions with no catalyst.

Another aspect is that I don't believe that a 1% CO₂ conversion is relevant in these conditions (highlighted in the title). Since the system is under CO₂ flow, this means that the available CO₂ for reduction regards to the saturation limit of their NaHCO₃ solution, i.e., 0.1 mol/L. This number is constant because they are introducing more CO₂ than consumed by the photoreduction. Therefore, they need to support this claim better - do the authors optimize this CO₂ flux by testing it in different CO₂ rates? In my opinion, rather than CO₂ conversion percentual, it would be more useful if they provide the quantum efficiency or at least an estimate of energy conversion (energy in produced CO/energy from irradiation).

Finally, I recommend standardizing the photocatalytic production to μmol/g.h, i.e., dividing the values by the amount of catalyst used. It is what is typically reported in the literature and helps to future comparison.

To the Reviewer: 1

General Comment: The manuscript by Pang et al. reported that the photocatalytic reduction of CO₂ over an Ag-modified Cr-decorated mixture of CaGa₄O₇/Ga₂O₃ and CaO exhibits an excellent CO formation with good selectivity, and the conversion of CO₂ in water can be up to 1% without any sacrificial agent. The materials are properly characterized, the number and quality of figures and tables is adequate. The paper is an important advance and of good quality, making it suitable for publication in Communications Chemistry. However, I believe there are some points that need to be addressed by the authors before acceptance.

Reply: Thank you very much for the kind comments. The authors revised our paper with the following reviewer's comments.

Comment 1-1: For comparison, please summarize some representative results of photocatalytic systems towards CO₂ reduction with H₂O in Table.

Reply 1-1: Thank you very much for your guidance. We have summarized some presentative results for the photocatalytic conversion of CO₂ by H₂O under the similar conditions, as shown in Supplementary Table 1. The corresponding sentence was added on page 5, line 12-13 highlighted in red in the main text as below.

The results reported in this study represent a significant breakthrough compared to previously published results, as summarized in **Supplementary Table 1**.

Comment 1-2: The Ca species on the Ga₂O₃ surface were in form of CaGa₄O₇ by XRD and XPS, while CaO was found by FTIR when the modification amount of Ca was low.

Please provide the Ca 2p XPS of CaO and the FTIR spectra of the CO₂-adsorbed on all Ga₂O₃_Ca_x samples and try to illustrate the existing form of Ca and what roles two Ca species play for the high conversion and selectivity during the photocatalytic CO₂ reduction.

Reply 1-2: Thank you for your suggestive comment. We have added the Ca 2p XPS of CaO in the Fig. 1C. The Ca 2p XPS peak of the Ga₂O₃_Ca sample is consistent with CaGa₄O₇, however, is different from that of CaO. We think that it is hard to confirm the formation of CaO using XPS due to the low amount of CaO.

We have already shown the FT-IR spectra of the CO₂-adsorbed on Ga₂O₃, CaO, Ga₂O₃_Ca_1.1, Ga₂O₃_Ca_3.3, and CaGa₄O₇ to confirm the formation of CaO in the Ga₂O₃_Ca_1.1 sample in the supporting information. As the reviewer 1 mentioned, the present result could clearly show the formation of CaO in Ga₂O₃_Ca_1.1. The FT-IR spectra were moved to the main text shown in Fig. 3 with necessary explanation on page 13-14 highlighted in red as below.

Fig. 3A shows the FTIR spectra of the CO₂-adsorbed samples after introducing CO₂ at ~0.2 Torr. When the Ga₂O₃ surface was modified with a small amount of Ca species, absorbance peaks attributable to CO₂ adsorption by both Ga₂O₃ and CaO were observed after CO₂ was introduced to the Ga₂O₃_Ca_1.1 sample.³⁴⁻³⁹ However, when the Ga₂O₃ surface was modified with large amounts of Ca species, the absorbance peaks attributed to CO₂ adsorption on Ga₂O₃ were observed to be of very-low intensity and to correspond mainly to the broad absorbance peaks derived from the adsorption of CO₂ on CaGa₄O₇. **Fig. 3B** shows the FTIR spectra of CO₂-adsorbed Ga₂O₃, Ga₂O₃_Ca_1.1, Ga₂O₃_Ca_3.3, and CaGa₄O₇ samples after introducing the same amount of CO₂ at various pressures in

the 0.1–40.0 Torr range. CO₂ was adsorbed significantly more on the Ga₂O₃_Ca_1.1 surface than on the Ga₂O₃ surface due to its adsorption at both Ga and Ca sites. However, the CaGa₄O₇ surface was not conducive to CO₂ adsorption; therefore, CO₂ adsorbed less onto the Ga₂O₃_Ca_3.3 surface than the Ga₂O₃_Ca_1.1 surface, as indicated by the OH deformation band [$\delta(\text{OH})$] at 1225 cm⁻¹, which is known to correspond to an intermediate species in the photocatalytic conversion of CO₂ by H₂O.⁴⁰

We further did some controlled experiment to explain the effect of CaO as shown in Fig. 4. The result indicates that the addition of CaO onto the Ga₂O₃ surface enhances the formation rate of CO and suppresses the formation of H₂ during the photocatalytic conversion of CO₂ by H₂O. In addition, the tight junction between the Ga₂O₃, CaO, and Ag@Cr cocatalyst is crucial to improve the photocatalytic activity and selectivity of the photocatalyst for the conversion of CO₂ into CO. The detail discussions were added on page 14, line 15-18, page 15, page 16, and page 17 line 1-6 highlighted in red in the main text as below.

In order to demonstrate that the presence of CaO on the Ga₂O₃ surface provides an environment conducive for enhancing photocatalytic activity and selectivity during the photocatalytic conversion of CO₂ into CO, we investigated photocatalytic performance during the conversion of CO₂ by H₂O over various Ag@Cr/CaO/Ga₂O₃ photocatalysts, the results of which are shown in **Fig. 4**. We found that the Ga₂O₃_Ca_1.1 photocatalyst (with a low amount of CaO generated on the Ga₂O₃ surface) significantly enhanced the rate of CO formation during the photocatalytic conversion of CO₂ by H₂O compared with bare Ga₂O₃ (**Fig. 4a and 4b**). However, no significant change in the rate of CO formation

and selectivity toward CO evolution was observed when 1.1 mol% CaO was physically loaded onto Ga₂O₃ by grinding (1.1 mol%CaO/Ga₂O₃) after which the Ag@Cr cocatalyst was loaded onto the CaO/Ga₂O₃ surface, when compared with those of bare Ga₂O₃ (**Fig. 4c**). Because uncalcined CaO-loaded Ga₂O₃ easily dissolves in H₂O, we increased the CaO loading on the Ga₂O₃ surface to 30 mol%, which resulted in an increase in the rate of CO formation and a decrease in H₂ formation using the Ag@Cr/30mol%CaO/Ga₂O₃ photocatalyst obtained using the same grinding method (**Fig. 4d**). However, no improvement in photocatalytic activity and selectivity during the conversion of CO₂ into CO by H₂O was observed when 30 mol% CaO was mixed with the prepared Ag@Cr/Ga₂O₃, even when ground together (**Fig. 4e**) or when mixed directly in the reaction solution (**Fig. 4f**). These results clearly reveal that the addition of CaO onto the Ga₂O₃ surface enhances the rate of CO formation and suppresses the formation of H₂ during the photocatalytic conversion of CO₂ by H₂O. In addition, the tight junction between Ga₂O₃, CaO, and the Ag@Cr cocatalyst is crucial for the superior photocatalytic activity and selectivity of the photocatalyst for the conversion of CO₂ into CO. In our previous work,⁴¹ we confirmed that Ag acts as an active site while the Cr(OH)₃·H₂O layer exterior to the Ag core increases CO₂ adsorption.⁴² Hence, the Ag@Cr cocatalyst should be loaded at the CaO/Ga₂O₃ interface to facilitate contact between the CaO-adsorbed CO₂ species and the Ag active sites.

Fig. 4. Formation rates and selectivities. Rates of formation of H₂ (blue bars), O₂ (green bars), and CO₂ (red bars), as well as selectivities toward CO evolution (black diamonds) for various photocatalysts: (a) Ga₂O₃, (b) Ga₂O₃_Ca_1.1, (c) Ga₂O₃ physically mixed with 1.1 mol% of CaO (with grinding), (d) Ga₂O₃ physically mixed with 30 mol% of CaO (with grinding), (e) Ag@Cr/Ga₂O₃ physically mixed with 30 mol% of CaO (with grinding), (f) Ga₂O₃ physically mixed with 30 mol% of CaO (without grinding) before being added into the reaction solution. Schematics of various photocatalysts are shown in (a)-(f). The photocatalysts in (a)-(e) were loaded with the Ag@Cr cocatalyst. Photocatalyst powder: 0.5 g, reaction solution volume: 1.0 L, additive: 0.1 M NaHCO₃, CO₂ flow rate: 30 mL min⁻¹, light source: 400-W high-pressure Hg lamp.

Comment 1-3: The specific compose of CaGa₄O₇/Ga₂O₃ photocatalyst physically mixed with 30 mol% of CaO with Ag-Cr as the cocatalyst should be provided to gain deeper

insights into the catalysis. 30 mol% of CaO are the optimum loading or not?

Reply 1-3: Thank you for your nice suggestion. Here the $\text{CaGa}_4\text{O}_7/\text{Ga}_2\text{O}_3$ is the sample $\text{Ca}_2\text{O}_3_Ca_3.3\text{mol}\%$. The effect of CaO amount is shown in Fig. 4. When 1.1 mol% of CaO was physically loaded on the Ga_2O_3 by grinding ($1.1\text{mol}\%\text{CaO}/\text{Ga}_2\text{O}_3$), there were no significant changes in the formation rate of CO and selectivity toward CO evolution compared with those for the bare Ga_2O_3 . Because CaO-loaded Ga_2O_3 without calcination can easily dissolve in H_2O , we increased the CaO loading on the Ga_2O_3 surface to 30 mol% and we observed that there was an increase in the formation rate of CO and a decrease in the formation of H_2 for the $\text{Ag-Cr}/30\text{mol}\%\text{CaO}/\text{Ga}_2\text{O}_3$ photocatalyst obtained using the same grinding method. However, further increasing the amount of CaO will not increase the photocatalytic activity, we think that the CaO layer on the Ga_2O_3 surface has reached saturation with loading amount of 30 mol%. The corresponding sentence was added on page 15, line 4-11 highlighted in red in the main text as below.

However, no significant change in the rate of CO formation and selectivity toward CO evolution was observed when 1.1 mol% CaO was physically loaded onto Ga_2O_3 by grinding ($1.1\text{mol}\%\text{CaO}/\text{Ga}_2\text{O}_3$) after which the Ag@Cr cocatalyst was loaded onto the $\text{CaO}/\text{Ga}_2\text{O}_3$ surface, when compared with those of bare Ga_2O_3 (**Fig. 4c**). Because uncalcined CaO-loaded Ga_2O_3 easily dissolves in H_2O , we increased the CaO loading on the Ga_2O_3 surface to 30 mol%, which resulted in an increase in the rate of CO formation and a decrease in H_2 formation using the $\text{Ag@Cr}/30\text{mol}\%\text{CaO}/\text{Ga}_2\text{O}_3$ photocatalyst obtained using the same grinding method (**Fig. 4d**).

Comment 1-4: Is it possible that H_2 and O_2 are generated from the over-all water

splitting?

Reply 1-4: All the H₂ generated is from the overall water splitting, however, the O₂ produced comes from two parts: overall water splitting and CO₂ reduction. As shown in Table 1, stoichiometric amounts of H₂ and CO as reduction products in addition to O₂ as the oxidation product (consumed $e^-/h^+ = 1$), which indicates that H₂O serves as the electron donor for the photocatalytic reduction of CO₂.

Comment 1-5: It's known that CaO reacts into Ca(OH)₂ in water, which is responsible for the high CO₂ adsorption. And I find that the formation rate of CO drops gradually in Fig.5(a). It means that the photocatalyst is not stable and/or the consumption of CaO is necessary for the photocatalytic CO₂ reduction. The authors should illustrate this clearly.

Reply 1-5: It is a very good question. In fact, in our previous work, we have found that the decrease of formation rate of CO is due to the photooxidation of Cr(OH)₃·H₂O.¹ The Cr(OH)₃·H₂O shell outside Ag particle can be oxidized to Cr⁶⁺ and dissolve into the solution during the photocatalytic conversion of CO₂. As shown in Supplementary Figure 2, the CO produced for the photocatalytic conversion of CO₂ after photoirradiation for 15 h over Ag@Cr/Ga₂O₃_Ca was more stable than that over Ag@Cr/Ga₂O₃. This indicates that the presence of Ca species is not only beneficial to improve the photocatalytic activity and selectivity, but also to improve the stability for the photocatalytic conversion of CO₂ to CO. With the CaO species around Ag active site, it can compensate for the reduced activity caused by the dissolution of Cr species, therefore, the photocatalytic stability was much higher on Ag@Cr/Ga₂O₃_Ca than Ag@Cr/Ga₂O₃. The corresponding sentence was added on page 7, line 6-11 and page 20, line 14-16, page 21, line 1-2.

Reference 1. Pang, R., Teramura, K., Asakura, H., Hosokawa, S. & Tanaka, T. Effect of

Cr Species on Photocatalytic Stability during the Conversion of CO₂ by H₂O. *J. Phys. Chem. C* **123**, 2894-2899 (2019).

Supplementary Figure 2. Formation rate of CO (red circle) and H₂ (blue triangle) for the photocatalytic conversion of CO₂ by H₂O over Ag@Cr/Ga₂O₃ (hollow mark) and Ag@Cr/Ga₂O₃_Ca (solid mark).

Page 7, line 6-11:

Additionally, CO production during the photocatalytic conversion of CO₂ after photoirradiation for 15 h over Ag@Cr/Ga₂O₃_Ca was more stable than that over Ag@Cr/Ga₂O₃ (**Supplementary Figure 2**), which indicates that the presence of Ca species is not only beneficial for improving photocatalytic activity and selectivity, but also for improving stability during the photocatalytic conversion of CO₂ to CO.

page 20, line 14-16, page 21, line 1-2:

Moreover, because the $\text{Cr}(\text{OH})_3 \cdot \text{H}_2\text{O}$ shell outside a Ag particle can be oxidized to Cr^{6+} and dissolved into the solution during the photocatalytic conversion of CO_2 ,⁴⁴ the presence of CaO around Ag active site compensates for the reduced activity resulting from the dissolution of Cr species; consequently, $\text{Ag}@ \text{Cr}/\text{Ga}_2\text{O}_3_ \text{Ca}$ is much more photocatalytically stable than $\text{Ag}@ \text{Cr}/\text{Ga}_2\text{O}_3$.

Comment 1-6: Energy band structure of $\text{CaGa}_4\text{O}_7/\text{Ga}_2\text{O}_3$ should be provided to clarify their justification for high efficiencies on photocatalytic CO_2 reduction.

Reply 1-6: Thank you so much for your suggestion. The energy band structures of Ga_2O_3 , $\text{CaGa}_4\text{O}_7/\text{Ga}_2\text{O}_3$, and CaGa_4O_7 were determined using the Mott-Schottky (Supplementary Figure 6) and Kubelka-Munk equations (Supplementary Figure 7). As shown in Supplementary Figure 8, both CB and VB of Ga_2O_3 are more positive than CaGa_4O_7 , therefore, it is possible to form a heterojunction system between Ga_2O_3 and CaGa_4O_7 , which will greatly improve the spatial separation efficiency of the photogenerated carriers. Thus, the $\text{CaGa}_4\text{O}_7/\text{Ga}_2\text{O}_3$ exhibited much higher photocatalytic efficiency than the bare Ga_2O_3 and CaGa_4O_7 . The corresponding sentences were added on page 17, line 12-18 and page 18, line 1-2 highlighted in red in the main text as below.

The Mott-Schottky plot (Supplementary Figure 6) and the absorption spectra converted from the diffuse reflectance spectra using the Kubelka-Munk equation (Supplementary Figure 7) enabled the conduction band (CB) and valence band (VB) positions of Ga_2O_3 , $\text{Ga}_2\text{O}_3_ \text{Ca}_0.62$, and CaGa_2O_7 to be determined, as shown in Supplementary Figure 8. Since the CB and VB of Ga_2O_3 are both more positive than those of CaGa_4O_7 , a heterojunction system between Ga_2O_3 and CaGa_4O_7 can be formed

that greatly improves the spatial separation efficiency of the photogenerated carriers.⁴³ Therefore, $\text{CaGa}_4\text{O}_7/\text{Ga}_2\text{O}_3$ exhibited a much higher photocatalytic efficiency than bare Ga_2O_3 and CaGa_4O_7 .

Supplementary Figure 6. Mott-Schottky plot for (A) $\text{Ga}_2\text{O}_3/\text{FTO}$, (B) $\text{Ga}_2\text{O}_3\text{-Ca}_{0.62}/\text{FTO}$, and (C) $\text{CaGa}_4\text{O}_7/\text{FTO}$ based on the results of the impedance measurements at a frequency of (a) 39.8, (b) 31.6, (c) 25.1 kHz. Electrolyte solution: Na_2SO_4 aq. (0.1 M, pH ca.7.0, Ag/AgCl), Atmosphere: N_2 .

Supplementary Figure 7. UV-visible spectra (A) and Davis-Mott plot presenting $(\alpha h\nu)^2$

versus photon energy (B) for the determination of band gap of Ga₂O₃ (black line), Ga₂O₃_Ca_0.62 (red line), and CaGa₄O₇ (blue line).

Supplementary Figure 8. Conduction band and valence band positions of Ga₂O₃, Ga₂O₃_Ca_0.62, and CaGa₄O₇.

Comment 1-7: A relevant work about CO₂ reduction using water as an electron source published recently may be referred: *Angew. Chem. Int. Ed.*, 2019, 58, 9491-9495.

Reply 1-7: Thank you for your kind suggestion. We added the paper as a reference (Ref. 8) to make the article more complete.

Reference:

8 Wu, L. Y. *et al.* Encapsulating perovskite quantum dots in iron-based metal-organic frameworks (MOFs) for efficient photocatalytic CO₂ reduction. *Angew. Chem. Int. Ed.* **58**, 9491-9495 (2019).

To the Reviewer: 2

General Comment: The motivation of the work was not clearly stated. Why did authors select a Ag-modified Cr-decorated mixture of CaGa₄O₇-loaded Ga₂O₃ and CaO photocatalyt? The statement should be clearly described in the paragraphs for introduction. Currently, it is hard to capture what is the motivation of the work, although the addition of Ca or CaO layer is the key to the works.

Reply: Thank you very much for the suggestive comment. Because the photocatalytic conversion of CO₂ over an excited semiconductor-based catalyst consists of three main steps: CO₂ adsorption, charge separation, and desorption of products, therefore, it can be deduced that the photocatalytic activity of the photocatalyst for CO₂ conversion can be improved by increasing the CO₂ adsorption, charge separation, and desorption of products. We have reported that modifying the photocatalyst surface with alkaline earth metals (e.g., Ca, Sr, and Ba) enhanced the conversion of CO₂ and the selectivity toward CO evolution. Moreover, we found that an Ag@Cr core/shell cocatalyst suppressed the backward reaction from CO and O₂ to CO₂, and enhanced the adsorption of CO₂, resulting in a highly selective photocatalytic CO₂ conversion. We want to exploit the above techniques to further increase the photocatalytic activity and selectivity for the conversion of CO₂ into CO, therefore we designed this Ag@Cr/Ga₂O₃_Ca photocatalyst. In order to make the motivation clearer, we revised the introduction part as shown on page 3, line 12-14, page 4, line 12-15 as shown in red in the main text.

Page 3, line 12-14:

In general, the photocatalytic conversion of CO₂ over an excited semiconductor-

based catalyst involves three main steps. First, CO₂ molecules are adsorbed on the photocatalyst surface.⁹⁻¹¹

Page 5, line 12-15:

Based on the processes involved in the photocatalytic conversion of CO₂ described previously, we deduce that the photocatalytic activity of the photocatalyst for CO₂ conversion can be improved by increasing CO₂ adsorption, charge separation, and product desorption.

Comment 2-1: To make readers understand the principle of CO₂ reduction, it is necessary to suggest the mechanism of charge transfer with schematic illustrations and a band diagram.

Reply 2-1: Thank you for your suggestive comment. We have proposed the possible mechanism for the photocatalytic conversion of CO₂ by H₂O over Ag@Cr/CaO/CaGa₂O₇/Ga₂O₃, as shown in Scheme 1. The CaO and CaGa₄O₇ formed on the Ga₂O₃ surface improve the adsorption of CO₂ on base sites and enhance the total photocatalytic efficiency, respectively. The following discussion was added on page 20 line 1-16 and page 21, line 1-2 highlighted in red in text as below.

Scheme 1. A plausible reaction mechanism. Schematic illustration of the mechanism

for the photocatalytic conversion of CO₂ into CO over Ag@Cr/CaO/CaGa₄O₇/Ga₂O₃.

Based on the above discussion, we propose a possible mechanism for the photocatalytic conversion of CO₂ by H₂O over Ag@Cr/CaO/CaGa₂O₇/Ga₂O₃, as shown in Scheme 1. During the photocatalytic conversion of CO₂ in an aqueous solution of NaHCO₃, the Cr(OH)₃·H₂O and CaO species that are in close contact with Ag particles easily form carbonate species,^{28,31} which greatly increases the concentration of CO₂-related species around the Ag active sites, thereby improving selectivity for the photocatalytic conversion of CO₂ into CO instead of water splitting. On the other hand, the Ga₂O₃/CaGa₄O₇ heterojunction improves the spatial separation efficiency of the photogenerated carriers, which increases the photocatalytic activity for the conversion of CO₂ into CO. Moreover, because the Cr(OH)₃·H₂O shell outside a Ag particle can be oxidized to Cr⁶⁺ and dissolved into the solution during the photocatalytic conversion of CO₂,⁴⁴ the presence of CaO around Ag active site compensates for the reduced activity resulting from the dissolution of Cr species; consequently, Ag@Cr/Ga₂O₃_Ca is much more photocatalytically stable than Ag@Cr/Ga₂O₃.

Comment 2-2: The amount of CO₂ adsorbed on each catalyst surface should be measured to confirm the enhanced CO₂ adsorption with the formation of CaO layer. The BET areas measured with N₂ gas decrease with increasing Ca but the CO₂ amount adsorbed should increase, which should be experimentally confirmed.

Reply 2-2: Thank you very much for the suggestive comment. We have measured the CO₂ adsorption over various Ca-modified-Ga₂O₃ using a CO₂ Temperature Programmed Desorption (TPD) machine, the result is shown as below:

Figure. R1 CO₂-TPD profiles of the photocatalysts.

Unfortunately, no significant CO₂ adsorption was detected in all the samples. This may be because the specific areas of Ga₂O₃ and Ca-modified-Ga₂O₃ are too small, it is difficult to detect the CO₂ adsorbed by the CO₂ TPD device we currently used.

Instead of TPD measurement, in this work, to confirm the enhanced CO₂ adsorption with the formation of CaO layer, we investigated the photocatalytic performance for the photocatalytic conversion of CO₂ by H₂O over various Ag@Cr/CaO/Ga₂O₃ photocatalysts. The result indicates that addition of CaO onto the Ga₂O₃ surface enhances the formation rate of CO and suppresses the formation of H₂ during the photocatalytic conversion of CO₂ by H₂O. The detail discussions were added on page 14, line 15-18, page 15, page 16, and page 17 line 1-6 highlighted in red in the main text as below.

In order to demonstrate that the presence of CaO on the Ga₂O₃ surface provides an environment conducive for enhancing photocatalytic activity and selectivity during the photocatalytic conversion of CO₂ into CO, we investigated photocatalytic performance

during the conversion of CO₂ by H₂O over various Ag@Cr/CaO/Ga₂O₃ photocatalysts, the results of which are shown in **Fig. 4**. We found that the Ga₂O₃_Ca_1.1 photocatalyst (with a low amount of CaO generated on the Ga₂O₃ surface) significantly enhanced the rate of CO formation during the photocatalytic conversion of CO₂ by H₂O compared with bare Ga₂O₃ (**Fig. 4a and 4b**). However, no significant change in the rate of CO formation and selectivity toward CO evolution was observed when 1.1 mol% CaO was physically loaded onto Ga₂O₃ by grinding (1.1 mol%CaO/Ga₂O₃) after which the Ag@Cr cocatalyst was loaded onto the CaO/Ga₂O₃ surface, when compared with those of bare Ga₂O₃ (**Fig. 4c**). Because uncalcined CaO-loaded Ga₂O₃ easily dissolves in H₂O, we increased the CaO loading on the Ga₂O₃ surface to 30 mol%, which resulted in an increase in the rate of CO formation and a decrease in H₂ formation using the Ag@Cr/30mol%CaO/Ga₂O₃ photocatalyst obtained using the same grinding method (**Fig. 4d**). However, no improvement in photocatalytic activity and selectivity during the conversion of CO₂ into CO by H₂O was observed when 30 mol% CaO was mixed with the prepared Ag@Cr/Ga₂O₃, even when ground together (**Fig. 4e**) or when mixed directly in the reaction solution (**Fig. 4f**). These results clearly reveal that the addition of CaO onto the Ga₂O₃ surface enhances the rate of CO formation and suppresses the formation of H₂ during the photocatalytic conversion of CO₂ by H₂O. In addition, the tight junction between Ga₂O₃, CaO, and the Ag@Cr cocatalyst is crucial for the superior photocatalytic activity and selectivity of the photocatalyst for the conversion of CO₂ into CO. In our previous work,⁴¹ we confirmed that Ag acts as an active site while the Cr(OH)₃·H₂O layer exterior to the Ag core increases CO₂ adsorption.⁴² Hence, the Ag@Cr cocatalyst should be loaded at the CaO/Ga₂O₃ interface to facilitate contact between the CaO-adsorbed CO₂ species and the Ag active sites.

Fig. 4. Formation rates and selectivities. Rates of formation of H₂ (blue bars), O₂ (green bars), and CO (red bars), as well as selectivities toward CO evolution (black diamonds) for various photocatalysts: (a) Ga₂O₃, (b) Ga₂O₃_Ca_1.1, (c) Ga₂O₃ physically mixed with 1.1 mol% of CaO (with grinding), (d) Ga₂O₃ physically mixed with 30 mol% of CaO (with grinding), (e) Ag@Cr/Ga₂O₃ physically mixed with 30 mol% of CaO (with grinding), (f) Ga₂O₃ physically mixed with 30 mol% of CaO (without grinding) before being added into the reaction solution. Schematics of various photocatalysts are shown in (a)-(f). The photocatalysts in (a)-(e) were loaded with the Ag@Cr cocatalyst. Photocatalyst powder: 0.5 g, reaction solution volume: 1.0 L, additive: 0.1 M NaHCO₃, CO₂ flow rate: 30 mL min⁻¹, light source: 400-W high-pressure Hg lamp.

To the Reviewer 3

General Comment: The paper from Pang et al describes a material -- Ag@Cr/Ga₂O₃_Ca -- with a remarkable CO₂ photoreduction activity to CO. The production of almost 1 mmol/g.h of CO is impressive, and it is higher than most of the recent papers about CO₂ reduction. The paper has an important feature: this is one of the only papers I've recently read that takes care of the oxidative reaction related to CO₂ reduction, i.e., O₂ evolution. In that sense, they could explore this data better.

Reply: Thank you very much for the suggestive comment. The authors revised our paper with the following reviewer's comments.

Comment 3-1: Fig 1 and Suppl Fig 1 show the production of CO, H₂, and O₂, comparatively, but it is hard to compare if the stoichiometry is correct or not. For instance, in Fig 1 for Ag@Cr/Ga₂O₃_Ca sample, it seems to me that CO production is around 0.8 mmol/h, H₂ is 0.2 mmol/h, and O₂ is 0.4 mmol/h. Since it is expected that CO₂  CO produces 1/2 O₂, as well as H₂O  H₂ + 1/2O₂, I would expect that the reaction would produce at least 0.5 mmol/h. Naturally, this indicates that probably this is a statistical variation question, but since the authors did not provide error bars for their experiments, it can not be confirmed.

Reply 3-1: Thank you for your comment. In order to make the data clearer, we have changed Figure 1 to Table 1. The details of the selectivity toward CO evolution and the balance between the consumed electrons and holes were clearly shown in Table 1. Stoichiometric amounts of H₂ and CO as reduction products in addition to O₂ as the oxidation product (consumed $e^-/h^+ = 1$), which indicates that H₂O serves as the electron

donor for the photocatalytic reduction of CO₂ instead of the other reducing agents.

Table 1 Photocatalytic conversion of CO₂ by H₂O vs. different photocatalysts.^[a]

Catalyst	Formation rates of products / $\mu\text{mol h}^{-1}$			Selec. toward CO (%)	Consumed e^-/h^+
	H ₂	O ₂	CO		
Bare Ga ₂ O ₃	240.9	122.8	9.8	4	1.02
Ag/Ga ₂ O ₃	248.3	172.7	102.1	29	1.01
Ag@Cr/Ga ₂ O ₃	148.5	316.4	499.6	77	1.02
Ag@Cr/Ga ₂ O ₃ _Ca	176.5	448.2	794.2	82	1.08

^[a] Photocatalyst powder: 0.5 g, reaction solution volume: 1.0 L, additive: 0.1 M NaHCO₃, CO₂ flow rate: 30 mL min⁻¹, light source: 400-W high-pressure Hg lamp.

Comment 3-2: A second question about their experiments is the power of the light source. They report using a 4,000 W high-pressure Hg lamp with a Quartz filter (line 268). It is significantly higher than typically reported in similar papers. The spectrum of these lamps includes deep-UV contributions, which can reduce CO₂ with no aid of catalysts. Please give more information about the lamp spectrum. See also JACS [dx.doi.org/10.1021/ja304930t](https://doi.org/10.1021/ja304930t), where the authors have discussed this reaction as well as the influence of a basic solid in increasing CO₂ reduction. Therefore, the authors should provide a "blank" experiment, i.e., the measurement of CO evolution in identical conditions with no catalyst.

Reply 3-2: Thank you for your kind suggestion. First of all, we are very sorry that we made a writing mistake, the power of the lamp is 400-W correctly instead of 4000-W.

Then, various controlled experiments were carried out to confirm the source of CO₂ are the introduced CO₂ gas during the photocatalytic conversion of CO₂ by H₂O, the results are shown in Supplementary Figure 3. There were no appreciable amounts of products detected in dark conditions or in the absence of a photocatalyst. In addition, H₂ was the main product formed when Ar gas was used instead of CO₂ or in the absence of NaHCO₃. The results obtained from the controlled experiments confirmed that the evolved CO originated from the gaseous CO₂ introduced to the samples and not from carbon contaminants. The following sentences were added on page 8, line 1-8 highlighted in red in the main text.

Supplementary Figure 3. Formation rates of H₂ (blue bars), O₂ (green bars), and CO (red bars) for the Ag-Cr/Ga₂O₃_Ca photocatalyst during photocatalytic conversion of CO₂. The data markers ○ and × indicate the presence and absence of each component,

respectively. Amount of photocatalyst: 0.5 g; Volume of reaction solution (H₂O): 1.0 L; Additive: 0.1 M NaHCO₃; CO₂ flow rate: 30 mL min⁻¹; Light source: 400-W high-pressure Hg lamp.

Various control experiments were carried out to confirm the source of CO during the photocatalytic conversion of CO₂ by H₂O, the results of which are shown in **Supplementary Figure 3**. We detected no appreciable amounts of products under dark conditions or in the absence of a photocatalyst. In addition, H₂ was the main product formed when Ar gas was used instead of CO₂ or in the absence of NaHCO₃. The control experiments confirmed that the evolved CO originated from the CO₂ gas introduced to the samples and not from carbon contaminants.

Comment 3-3: Another aspect is that I don't believe that a 1% CO₂ conversion is relevant in these conditions (highlighted in the title). Since the system is under CO₂ flow, this means that the available CO₂ for reduction regards to the saturation limit of their NaHCO₃ solution, i.e., 0.1 mol/L. This number is constant because they are introducing more CO₂ than consumed by the photoreduction. Therefore, they need to support this claim better - do the authors optimize this CO₂ flux by testing it in different CO₂ rates? In my opinion, rather than CO₂ conversion percentual, it would be more useful if they provide the quantum efficiency or at least an estimate of energy conversion (energy in produced CO/energy from irradiation).

Reply 3-3: Thank you for your suggestive comment. As you mentioned, we also think that it is very important to investigate dependence of activity and selectivity on the

concentration of CO₂ in an aqueous solution. Frankly, we have already discussed the effect of CO₂ partial pressure in our previous work (*J. Phys. Chem. C*, **2017**, *121*, 8711-8721). It was reported that the formation rate of CO increased remarkably with the partial pressure of CO₂, which indicates that the formation rate of CO is dependent on the solubility of CO₂ in water depending on the partial pressure (so-called Henry's law). In addition, the formation rate of CO depends on the concentration of NaHCO₃. In this study, we carried out all the reactions under the optimized reaction condition in the basis of this previous work. On the other hand, we accept what the reviewer 3 said that he/she does not believe that a 1% CO₂ conversion is relevant in these conditions (highlighted in the title). In order to make the theme of the article clearer, we changed the title to "Remarkable Enhancement of CO Evolution in Ca-Modified Photocatalytic Conversion of CO₂ by H₂O".

Regarding the quantum efficiency (QE) the reviewer 3 mentioned, we have already tried to measure the QE under light irradiation at 254 nm which is one of emission lines of high-pressure Hg lump (254, 365, 404, 435, 546, 577, and 579 nm) we used, because our photocatalysts are activated at less than 300 nm of wavelength. The energy band structures of Ga₂O₃, CaGa₄O₇/Ga₂O₃, and CaGa₄O₇ were determined using the Mott-Schottky (Supplementary Figure 6) and Kubelka-Munk equations (Supplementary Figure 7) to explain the role of CaGa₄O₇. The corresponding sentences were added on page 17, line 12-18 and page 18, line 1-2 highlighted in red in the main text. As you well know, it was difficult to obtain photon flux at less than 300 nm; therefore, the QE of CO evolution was unstable, and the estimation was very low in this work. We really understand that it is very important to show QE value in the field of photocatalysis such as overall water

splitting. The QE was less than 1% at 365 nm by using Xe lamp in the case of Ag/Al-SrTiO₃ which we recently reported (*ACS Applied Energy Materials*, 2020, 3 1468-1475). As it now stands, we have the technique to measure the QE at 365 nm, however, do not have at 254 nm as mentioned above. Thus, we are right now fabricating Ga₂O₃ photocatalysts which can absorb much higher wavelength than 300 nm (*J. Phys. Chem. C*, **2018**, *122*, 21132-21139). We will show the QE constantly in the field of CO₂ photoreduction in the future in the same manner as overall water splitting.

Comment 3-4: Finally, I recommend standardizing the photocatalytic production to $\mu\text{mol/g.h}$, i.e., dividing the values by the amount of catalyst used. It is what is typically reported in the literature and helps to future comparison.

Reply 3-4: Thank you for your comment. As we known, the definition of the chemical reaction rate is $\bar{v} = -\Delta c/\Delta t$ (\bar{v} is average reaction rate; Δc is the concentration of products; Δt is the reaction time), and the function of the catalyst is to reduce the active energy in the reaction. Thus, the catalytic performance does not depend on the weight of the catalyst. Typically, the formation rate of product is proportional to amount of catalyst when it is small, and then the formation rate of product becomes stable (it does not depend on amount of catalyst), which means that rate-determining step should be changed. Generally, we show the stable formation rate to compare each activity in the research field of catalysts and catalysis. In fact, we found that the formation rate of CO over 0.3 g catalyst was similar with that over 0.5g catalyst. Here we used 0.5 g of catalyst, mainly for comparison with our previous papers. In addition, we have gone on using the unit of $\mu\text{mol h}^{-1}$ in the previous papers (approximately 10 papers). Therefore, we believe that using $\mu\text{mol h}^{-1}$ is a more accurate expression than $\mu\text{mol g}^{-1} \text{h}^{-1}$.

Reviewers' comments:

Reviewer #1 (Remarks to the Author):

This paper synthesized Ag@Cr/CaO/CaGa4O7/Ga2O3 and obtained a satisfied activity in photocatalytic CO2 reduction to CO. A detailed investigation was performed to reveal the nature behind the photocatalytic performance. However, in my opinion, some important issues are not resolved.

My comments are listed below:

1. The title should be modified. Ca was introduced to modify a photocatalyst. Additionally, it would be better that the information of the synthesized photocatalyst appears in the title.
2. The most serious issue is if the CaO exists in the composite photocatalyst? The authors think the CO2 adsorption ability of CaO acts an important role in photocatalytic reduction CO2 to CO. However, there is no powerful proof to verify the existence of CaO species. The XRD and XPS analysis proves that the Ca signal comes from CaGa4O7. Additionally, it can be noted that the Ca content is significantly lower than Ga. I cannot understand why some Ca react with Ga2O3 to form CaGa4O7, whereas some Ca stay in the form of CaO? Third, CaO is unstable in the presence of water or CO2, and easily forms Ca(OH)2 or CaCO3. On the contrary, the reverse process (release CO2) is difficult to be performed at room temperature. How CaO can continuously adsorb and desorb CO2 in photocatalytic reactions?
3. Fig. 1c, the "Ca2p" should be added to make readers understand it easily.
4. Page 9, line 13-14, the peak change indicates that the amount of Ca plays a role in altering the morphology of Ga2O3. I don't think so. Actually, the SEM analysis does not show that the morphology of Ga2O3 change a lot. The change in XRD peak can be ascribed to the formed CaGa4O7 species.
5. Fig. 3, IR spectra was performed to confirm the CO2 adsorption performance of the composite photocatalyst. It is noted that pure CaGa4O7 shows poor ability in CO2 adsorption. What's the BET surface area of CaGa4O7? It would greatly affect the adsorption performance. Additionally, what's the photocatalytic activity of pure CaGa4O7? Additionally, the appeared adsorption peak in Fig. 3 are not assigned. It can be noted that the adsorption peak of the composite is more similar to that of CaGa4O7, not CaO. So, it is possible that the state of CaGa4O7 in the composite is different from bulk CaGa4O7, which endow the composite good capability in CO2 adsorption.
6. Page 14, line 10, the peak at 1225 cm⁻¹ was ascribed to an intermediate species in the photocatalytic conversion of CO2 by H2O. It can be noted that only pure CaO does not shows the peak. It can be an indirectly proof that CaO may not exist in the composite.
7. Fig. 4 caption, the label of a, b, c, d, f is not consist with the presented scheme. Based on the scheme of the photocatalysts, Ag@Cr co-catalyst is also included. What's the difference of the sample d and e?
8. Page 19, Fig. 5, authors think that 12CO comes from NaHCO3. It can be noted the 12CO generation rate reaches about 100-200 μmol h⁻¹. In Fig. S3, In Fig. S3, although the NaHCO3 is still included, the CO generation rate can be ignored in the absence of CO2. Why?

Reviewer #2 (Remarks to the Author):

The manuscript has been substantially modified, responding to reviewer's comments. I think that it is acceptable now.

Reviewer #3 (Remarks to the Author):

In this revision, the authors have addressed my main concerns regarding the 1st version. In my previous review, I've highlighted that this paper is better than I'm frequently reading, and I

believe that the current manuscript version is acceptable for publication in its present form.

However, I'd rather insist on some points for future discussions:

- In Fig 1 (changed to Table 1), the authors didn't provide statistical relevance. This is a critical point in catalysis since we observe many outliers in CO₂ reduction literature. Therefore, this aspect is still missing in the paper. I agree that the product yields do not open a discussion about the feasibility (they're in fact much higher than typically reported), but the differences among Ag@Cr/Ga₂O₃ and Ag@Cr/Ga₂O₃_Ca might be not significant, depending on the error bars.
- The standardization of $\mu\text{mol}\cdot\text{g}^{-1}\cdot\text{h}^{-1}$ is adopted by the most of the authors to compare the results from different sources. I understand the choice of the authors but they didn't provide the as-mentioned study comparing 0.3 and 0.5 g of catalyst. This is an open discussion: what is the reference for the catalytic activity? The standardization using g of catalyst is indirectly saying the available surface area, which is actually related to the total number of available catalytic sites. However, this aspect is not essential for the comprehension of the paper and it is more a matter for future discussions.

To the Reviewer: 1

Comment 1-1: The title should be modified. Ca was introduced to modify a photocatalyst. Additionally, it would be better that the information of the synthesized photocatalyst appears in the title.

Reply 1-1: Thank you very much for your suggestion. We have modified the title for clarity by adding the specific name of the photocatalyst:

Old title: "Remarkable Enhancement of CO Evolution in Ca-Modified Photocatalytic Conversion of CO₂ by H₂O"

New title: "Remarkably Enhanced CO Evolution for Photocatalytic Conversion of CO₂ by H₂O over Ca Species-Modified Ga₂O₃"

Comment 1-2: The most serious issue is if the CaO exists in the composite photocatalyst? The authors think the CO₂ adsorption ability of CaO acts an important role in photocatalytic reduction CO₂ to CO. However, there is no powerful proof to verify the existence of CaO species. The XRD and XPS analysis proves that the Ca signal comes from CaGa₄O₇. Additionally, it can be noted that the Ca content is significantly lower than Ga. I cannot understand why some Ca react with Ga₂O₃ to form CaGa₄O₇, whereas some Ca stay in the form of CaO? Third, CaO is unstable in the presence of water or CO₂, and easily forms Ca(OH)₂ or CaCO₃. On the contrary, the reverse process (release CO₂) is difficult to be performed at room temperature. How CaO can continuously adsorb and desorb CO₂ in photocatalytic reactions?

Reply 1-2: We appreciate the above suggestion. We are in full agreement with the reviewer's arguments that CaO is destabilized in the presence of H₂O and CO₂, and other Ca species such as Ca(OH)₂ and CaCO₃ should be formed. Although some Ca

species such as CaCO_3 are stable at room temperature, species such as Ca(OH)_2 and CaCO_3 can co-exist at equilibrium in the presence of CO_2 in an NaHCO_3 aqueous solution.^{1,2} For example, it has been reported that in the pH range of 5–10, CaOHCO_3^- is the main species on the surface of calcite.³ In this work, the pH during the photocatalytic reaction is approximately 6.8. Moreover, in our previous work we found that basic oxides and hydroxides such as Cr(OH)_3 ,⁴ SrO ,⁵ and rare earth (RE) hydrates and oxides⁶ function as good CO_2 storage materials by forming the corresponding (hydroxy)carbonate compounds (e.g. $\text{Cr(OH)}_x(\text{CO}_3)_y$ and $\text{RE}_2(\text{OH})_{2(3-x)}(\text{CO}_3)_x$) and improve the activity and selectivity toward CO evolution during the photocatalytic reaction in a NaHCO_3 aqueous solution with flowing CO_2 . In the current system, we expect that amorphous CaO is generated in the as-syn photocatalysts, while calcium (hydroxy)carbonate and/or carbonate hydrate (here, we named the possible Ca species as $\text{Ca(OH)}_x(\text{CO}_3)_y$) are working states during the reaction.

On the other hand, we have already concluded that CaGa_4O_7 , which was detected by XRD and XPS, does not contribute toward the photocatalytic conversion of CO_2 by H_2O . As shown in Supplementary Figure 1h, only overall water splitting proceeded over Ag-modified CaGa_4O_7 even in the presence of CO_2 . Actually, the formation rate of CO decreased with increasing loading amount of Ca whereas that of H_2 increased at the same time in Supplementary Figure 1, because CaGa_4O_7 was generated. Unfortunately, it was barely detected CaO by XRD and XPS due to the low concentration of Ca species. To obtain indirect evidence, we have already measured the FTIR spectra of CO_2 species adsorbed on these photocatalysts and references. First, the spectrum on the $\text{Ga}_2\text{O}_3\text{-Ca}_1.1$ surface was different from that on $\text{Ga}_2\text{O}_3\text{-Ca}_3.3$. Second, the spectra on $\text{Ga}_2\text{O}_3\text{-Ca}_1.1$ and $\text{Ga}_2\text{O}_3\text{-Ca}_3.3$, in which CaGa_4O_7 was

detected, were similar to those on CaO and CaGa₄O₇, respectively. These results led to the same conclusion mentioned above, that amorphous CaO is generated in the as-syn photocatalysts.

As the reviewer pointed out, we have no strong proof for the existence of CaO species except for the FTIR spectra. Therefore, we are willing to change all such instances in the text to “Ca” or “Ca species”, based on the reviewer’s strong recommendation. On the other hand, we ourselves prefer the term “CaO”, if that is acceptable. In the field of catalysis, elements like Ca should exist as oxides (CaO) after calcination when there is no evidence for forming mixed oxides.

We added the above discussion in the part pertaining to the reaction mechanism on page 22 (red-colored text). Additionally, in Scheme 1, the expression “CaO·CO₂” was changed to calcium (hydroxy)carbonate (Ca(OH)_x(CO₃)_y) to denote the possible Ca species in solution.

When we wrote the paper, we had planned to measure the Ca K-edge XAFS spectra as possible direct evidence of CaO. Unfortunately, that turned out to be impossible for two reasons. One is the COVID-19 pandemic, which led to the closure of almost all synchrotron radiation facilities in Japan. Although we had been assigned extraordinary beamtime a couple of times, we were not able to perform the measurement. The other issue is the very low absorption energy of Ca K-edge XAFS spectrum (4038.5 eV), which necessitates a special measurement technique according to our discussion with the technical staff at the synchrotron radiation facilities. Thus, we are afraid that direct evidence for CaO will take quite some time to obtain.

Comment 1-3: Fig. 1c, the “Ca2p” should be added to make readers understand it

easily.

Reply 1-3: Thank you for your suggestion. We added the label “Ca 2p” in Fig. 1C to facilitate understanding.

Comment 1-4: Page 9, line 13-14, the peak change indicates that the amount of Ca plays a role in altering the morphology of Ga₂O₃. I don't think so. Actually, the SEM analysis does not show that the morphology of Ga₂O₃ change a lot. The change in XRD peak can be ascribed to the formed CaGa₄O₇ species.

Reply 1-4: As the reviewer mentioned, the SEM images of various Ga₂O₃_Ca photocatalysts were almost all the same. On the other hand, their XRD patterns were not different from that of Ga₂O₃, especially when the amount of Ca was lower than 1.1 mol%. Thus, we could assume that the morphologies of Ga₂O₃_Ca photocatalysts were quite similar to that of Ga₂O₃. When the amount of Ca was higher than 1.1 mol%, Ga₂O₃ particles in the modified photocatalysts became larger and sharper compared to Ga₂O₃_Ca at a low amount of Ca. This change could be caused by two possible reasons: surface modification by Ca species, and the formation of CaGa₄O₇ on Ga₂O₃. Based on the XPS data and SEM images, the latter reason is more likely. We revised this part in the main text (page 9, line 17–18 and page 10, line 1–2, in red color) as follows:

However, there was a clear increase in the peak intensity at $2\theta = 30.1^\circ$ and an apparent decrease in that at $2\theta = 30.5^\circ$ with increasing amount of Ca species (**Fig. 1B**), which are possibly ascribed to the formation of CaGa₄O₇ species on Ga₂O₃.

Comment 1-5: Fig. 3, IR spectra was performed to confirm the CO₂ adsorption performance of the composite photocatalyst. It is noted that pure CaGa₄O₇ shows poor

ability in CO₂ adsorption. What's the BET surface area of CaGa₄O₇? It would greatly affect the adsorption performance. Additionally, what's the photocatalytic activity of pure CaGa₄O₇? Additionally, the appeared adsorption peak in Fig. 3 are not assigned. It can be noted that the adsorption peak of the composite is more similar to that of CaGa₄O₇, not CaO. So, it is possible that the state of CaGa₄O₇ in the composite is different from bulk CaGa₄O₇, which endow the composite good capability in CO₂ adsorption.

Reply 1-5: The BET surface area of CaGa₄O₇ particles was approximately 6.8 m² g⁻¹, smaller than those of the Ga₂O₃_Ca photocatalysts. As shown in the FTIR spectra (Fig. 3), bare CaGa₄O₇ had poor capability for adsorbing CO₂. Moreover, when we carried out the photocatalytic conversion of CO₂ by H₂O over Ag-Cr/CaGa₄O₇, only overall water splitting proceeded even in the presence of CO₂ as shown in Supplementary Figure 1h. This result indicates that bare CaGa₄O₇ is unfavorable for the photocatalytic conversion of CO₂ in an aqueous solution. On the other hand, the Ga₂O₃_Ca_3.3 photocatalyst, which mainly consisted of Ga₂O₃ particles covered by CaGa₄O₇, had good capability for CO₂ adsorption (Fig. 3d). Thus, we agree with the reviewer's opinion that the CaGa₄O₇ in the composite would be different from bulk CaGa₄O₇. Unfortunately, the CaGa₄O₇ in the composite also showed poor selectivity toward CO evolution. As shown in Supplementary Figure 1a–1g, the formation rate of H₂ over the Ag-Cr/Ga₂O₃_Ca_x samples increased monotonically with increasing amount of CaGa₄O₇ on the Ga₂O₃ surface. We surmise that the improved rate of CO formation and selectivity toward CO evolution over Ga₂O₃_Ca photocatalysts are not due to the adsorption performance of CaGa₄O₇ on the surface. In fact, the FTIR peaks of CO₂ adsorbed on Ga₂O₃_Ca_1.1, which were assigned to carbonate and bicarbonate species,

were almost consistent with those on CaO but not with that of CaGa₄O₇, as shown in Fig. 3b and 3c. Therefore, we attribute the high formation rate of CO and good selectivity toward CO evolution over Ga₂O₃_Ca with a low amount of Ca to modification of the Ga₂O₃ surface by the other Ca species such as CaO rather than CaGa₄O₇. According to the reviewer's comment, the peaks were assigned in the main text on page 13, line 10–13 and page 14, line 1–6 (colored in red) as follows:

When CO₂ was introduced into the Ga₂O₃ sample, three absorbance peaks were observed at 1634, 1432, and 1225 cm⁻¹, which can be ascribed to asymmetric CO₃ stretching vibrations [$\nu_{as}(\text{CO}_3)$], symmetric CO₃ stretching vibrations [$\nu_s(\text{CO}_3)$] of monodentate bicarbonate species (m-HCO₃-Ga), and OH deformation vibrations [$\delta(\text{OH})$], respectively.⁷⁻⁹ The absorbance peaks at 1699 and 1636 cm⁻¹ for the CO₂-adsorbed CaO sample can be attributed to bridging carbonate stretching and asymmetric CO₃ stretching vibrations [$\nu_{as}(\text{CO}_3)$] of the bicarbonate species, respectively. The broad structureless absorbance peaks between 1480 and 1318 cm⁻¹ can be attributed to the symmetric and asymmetric CO₃ stretching of unidentate carbonate, as well as the symmetric CO₃ stretching [$\nu_s(\text{CO}_3)$] of bicarbonate.¹⁰⁻¹⁴

Comment 1-6: Page 14, line 10, the peak at 1225 cm⁻¹ was ascribed to an intermediate species in the photocatalytic conversion of CO₂ by H₂O. It can be noted that only pure CaO does not show the peak. It can be an indirectly proof that CaO may not exist in the composite.

Reply 1-6: The peak at 1225 cm⁻¹ is assigned to the OH deformation vibration band [$\delta(\text{OH})$] on Ga [$\delta(\text{OH})$ -Ga], which has been known as an intermediate species in the photocatalytic conversion of CO₂ by H₂O over Ga₂O₃.¹⁵ As shown in Fig. 4, we found

that the modification of Ca species further increased the amount of intermediate [$\nu_s(\text{CO}_3)\text{-Ca}$] (at 1408 cm^{-1}) for the photocatalytic conversion of CO_2 , in addition to the presence of intermediate species on the Ga_2O_3 surface [$\delta(\text{OH})\text{-Ga}$]. Compared with CO_2 adsorption on CaO , the adsorption ascribed to $\nu_s(\text{CO}_3)\text{-Ca}$ on CaO was the same as that on $\text{Ga}_2\text{O}_3\text{-Ca}_{1.1}$. Therefore, we concluded that CaO should exist on the $\text{Ga}_2\text{O}_3\text{-Ca}_{1.1}$ surface.

Comment 1-7: Fig. 4 caption, the label of a, b, c, d, f is not consistent with the presented scheme. Based on the scheme of the photocatalysts, Ag@Cr co-catalyst is also included. What's the difference of the sample d and e?

Reply 1-7: We apologize for the possible confusion. Sample d was prepared by first physically mixing Ga_2O_3 and CaO by grinding, and then loading the Ag@Cr cocatalyst. Sample e was mixed by grinding the prepared $\text{Ag@Cr Ga}_2\text{O}_3$ and CaO together. This difference is now highlighted in red in page 17.

Further, the photocatalysts used in different panels in Fig. 5 (Fig. 4 in the previous version) are now clearly labeled in the caption as follows.

Fig. 5. Product formation rates and selectivities. Rates of formation of H₂ (blue bars), O₂ (green bars), and CO (red bars), as well as selectivities toward CO evolution (black diamonds) for various photocatalysts: (a) Ag@Cr/Ga₂O₃; (b) Ag@Cr/Ga₂O₃_Ca_1.1; (c) Ag@Cr/(1.1mol%CaO/Ga₂O₃), namely Ga₂O₃ physically mixed with 1.1 mol% of CaO by grinding before loading the Ag@Cr cocatalyst; (d) Ag@Cr/(30mol%CaO/Ga₂O₃), which is similar to (c) except for using 30 mol% of CaO; (e) Ag@Cr/Ga₂O₃ + 30mol%CaO, namely Ag@Cr/Ga₂O₃ physically mixed with 30 mol% of CaO by grinding; and (f) Ag@Cr/Ga₂O₃ and 30 mol% of CaO without mixing before adding into the reaction solution. Schematic structures of the photocatalysts are shown at bottom. Photocatalyst powder: 0.5 g, reaction solution volume: 1.0 L, additive: 0.1 M NaHCO₃, CO₂ flow rate: 30 mL min⁻¹, light source: 400 W high-pressure Hg lamp.

Comment 1-8: Page 19, Fig. 5, authors think that ¹²CO comes from NaHCO₃. It can be noted the ¹²CO generation rate reaches about 100-200 μmol h⁻¹. In Fig. S3, although the NaHCO₃ is still included, the CO generation rate can be ignored in the absence of CO₂.

Why?

Reply 1-8: CO₂ gas is undoubtedly water-soluble, and hydrated CO₂ (CO₂(aq)) reacts with H₂O to form carbonic acid (H₂CO₃). The dissociation of H₂CO₃ produces bicarbonate (HCO₃⁻) and carbonate ions (CO₃²⁻) in two stages depending on the pH value. These four reactions reach equilibrium during the photocatalytic reactions:

In Supplementary Figure S3, Ar gas was flowed instead of CO₂ in an aqueous solution of NaHCO₃. The solution pH was approximately 8.8. On the other hand, in Fig. 5, CO₂ was bubbled in the solution with a pH of approximately 6.7. As shown in Fig. R1 below, the concentrations of CO₂(aq), HCO₃⁻, and CO₃²⁻ in a 0.10 M aqueous solution of NaHCO₃ are completely different between pH 8.8 and 6.7. In our previous work, CO₂(aq) was proven as the direct reactant for the photocatalytic conversion of CO₂ with H₂O as an electron donor.¹⁵ Due to the higher concentration of CO₂(aq) at pH 6.6 than pH 8.8, the formation rate of CO in Fig. 5 was much higher than that in Supplementary Figure S3.

Fig. R1. Calculated pH-dependent concentrations of CO₂(aq) (circle), HCO₃⁻ (triangle), and CO₃²⁻ (square) in a 0.10 M aqueous solution of NaHCO₃ at 303 K under 101.325 kPa of CO₂.

References

- 1 Li, J. & Wang, T. On the deactivation of alkali solid catalysts for the synthesis of glycerol carbonate from glycerol and dimethyl carbonate. *Reac. Kinet. Mech. Cat.* **102**, 113-126 (2011).
- 2 Liu, Z. & Dreybrod, W. Dissolution kinetics of calcium carbonate minerals in H₂O CO₂ solutions in turbulent flow: The role of the diffusion boundary layer and the slow reaction H₂O+ CO₂→ H⁺+ HCO₃⁻. *Geochim. Cosmochim. Acta* **61**, 2879-2889 (1997).
- 3 Dolgaleva, I., Gorichev, I., Izotov, A. & Stepanov, V. Modeling of the effect of pH on the calcite dissolution kinetics. *Theor. Found. Chem. Eng.* **39**, 614-621 (2005).

- 4 Pang, R., Teramura, K., Asakura, H., Hosokawa, S. & Tanaka, T. Effect of thickness of chromium hydroxide layer on Ag cocatalyst surface for highly selective photocatalytic conversion of CO₂ by H₂O. *ACS Sustain. Chem. Eng.* **7**, 2083-2090 (2018).
- 5 Yoshizawa, S. *et al.* Important role of strontium atom on the surface of Sr₂KTa₅O₁₅ with a tetragonal tungsten bronze structure to improve adsorption of CO₂ for photocatalytic conversion of CO₂ by H₂O. *ACS Appl. Mater. Interfaces* **11**, 37875-37884 (2019).
- 6 Huang, Z., Teramura, K., Asakura, H., Hosokawa, S. & Tanaka, T. CO₂ capture, storage, and conversion using a praseodymium-modified Ga₂O₃ photocatalyst. *J. Mater. Chem. A* **5**, 19351-19357 (2017).
- 7 Tsuneoka, H., Teramura, K., Shishido, T. & Tanaka, T. Adsorbed species of CO₂ and H₂ on Ga₂O₃ for the photocatalytic reduction of CO₂. *J. Phys. Chem. C* **114**, 8892-8898 (2010).
- 8 Collins, S. E., Baltanás, M. A. & Bonivardi, A. L. An infrared study of the intermediates of methanol synthesis from carbon dioxide over Pd/β-Ga₂O₃. *J. Catal.* **226**, 410-421 (2004).
- 9 Collins, S. E., Baltanás, M. A. & Bonivardi, A. L. Infrared spectroscopic study of the carbon dioxide adsorption on the surface of Ga₂O₃ polymorphs. *J. Phys. Chem. B* **110**, 5498-5507 (2006).
- 10 Fukuda, Y. & Tanabe, K. Infrared study of carbon dioxide adsorbed on magnesium and calcium oxides. *Bull. Chem. Soc. Jpn.* **46**, 1616-1619 (1973).
- 11 Philipp, R. & Fujimoto, K. FTIR spectroscopic study of carbon dioxide adsorption/desorption on magnesia/calcium oxide catalysts. *J. Phys. Chem.* **96**,

9035-9038 (1992).

- 12 Busca, G. & Lorenzelli, V. Infrared spectroscopic identification of species arising from reactive adsorption of carbon oxides on metal oxide surfaces. *Mater. Chem.* **7**, 89-126 (1982).
- 13 Tsyganenko, A. & Filimonov, V. Infrared spectra of surface hydroxyl groups and crystalline structure of oxides. *J. Mol. Struct.* **19**, 579-589 (1973).
- 14 Busca, G., Lamotte, J., Lavalley, J. C. & Lorenzelli, V. FT-IR study of the adsorption and transformation of formaldehyde on oxide surfaces. *J. Am. Chem. Soc.* **109**, 5197-5202 (1987).
- 15 Teramura, K. *et al.* Which is an intermediate species for photocatalytic conversion of CO₂ by H₂O as the electron donor: CO₂ molecule, carbonic acid, bicarbonate, or carbonate ions? *J. Phys. Chem. C* **121**, 8711-8721 (2017).

To the Reviewer: 2

The manuscript has been substantially modified, responding to reviewer's comments. I think that it is acceptable now.

Reply: Thank you very much for your recommendation to accept our revised manuscript.

To the Reviewer: 3

In this revision, the authors have addressed my main concerns regarding the 1st version. In my previous review, I've highlighted that this paper is better than I'm frequently reading, and I believe that the current manuscript version is acceptable for publication in its present form. However, I'd rather insist on some points for future discussions:

Comment 3-1: In Fig 1 (changed to Table 1), the authors didn't provide statistical relevance. This is a critical point in catalysis since we observe many outliers in CO₂ reduction literature. Therefore, this aspect is still missing in the paper. I agree that the product yields do not open a discussion about the feasibility (they're in fact much higher than typically reported), but the differences among Ag@Cr/Ga₂O₃ and Ag@Cr/Ga₂O₃_Ca might be not significant, depending on the error bars.

Reply 3-1: Thank you for your kind comments. We have carried out the photocatalytic reaction for the conversion of CO₂ by H₂O over Ag@Cr/Ga₂O₃ and Ag@Cr/Ga₂O₃_Ca for at least four times, and the errors of the product formation rates (H₂, O₂, and CO) were smaller than 5%. We have mentioned this information in page 7, line 1–4 (colored in red).

Comment 3-2. The standardization of $\mu\text{mol} \cdot \text{g}^{-1} \cdot \text{h}^{-1}$ is adopted by the most of the authors to compare the results from different sources. I understand the choice of the authors but they didn't provide the as-mentioned study comparing 0.3 and 0.5 g of catalyst. This is an open discussion: what is the reference for the catalytic activity? The standardization using g of catalyst is indirectly saying the available surface area, which is actually related to the total number of available catalytic sites. However, this aspect is not essential for the

comprehension of the paper and it is more a matter for future discussions.

Reply 3-2: We totally agree with the reviewer's comment that the catalytic activity per unit area is an important indicator for evaluating the photocatalyst performance. Actually, qualitative methods do not lead to accurate conclusions. We also think that the catalytic activity should be compared quantitatively. As the reviewer mentioned, more detailed experiments should be required in the next step, such as evaluating the dependence of photocatalytic activity on the specific surface area. We really appreciate the reviewer's comment and will take care of amount of active sites to discuss the catalytic activity and selectivity in the next papers.

REVIEWERS' COMMENTS:

Reviewer #1 (Remarks to the Author):

The revised version is publishable. It addressed this reviewer's comments made in the 1st review.

Reviewer #3 (Remarks to the Author):

Considering that my questions about this manuscript were minimal in the last version, I agree with the paper in its current form. My major concern (statistical representativity) has been addressed by their confirmation of 4 replicates with less than 5% deviation. Based on their information, I don't have further comments about this paper.

To the Reviewer: 1

The revised version is publishable. It addressed this reviewer's comments made in the 1st review.

Reply: Thank you very much for accepting our revised manuscript.

To the Reviewer: 3

Considering that my questions about this manuscript were minimal in the last version, I agree with the paper in its current form. My major concern (statistical representativity) has been addressed by their confirmation of 4 replicates with less than 5% deviation. Based on their information, I don't have further comments about this paper.

Reply: Thank you very much for your recommendation to accept our revised manuscript.